# Structural and dynamic insights into the biased signaling mechanism of the human kappa opioid receptor

Chiyo Suno-Ikeda[1,14], Ryo Nishikawa[2,14], Riko Suzuki [3,14], Shun Yokoi [4,5], Seiya Iwata[2], Tomoyo Takai[1], Takaya Ogura[6], Mika Hirose[7], Akihisa Tokuda[5], Risako Katamoto[5], Akitoshi Inoue[1], Eri Asai[1], Ryoji Kise [3], Yukihiko Sugita [8,9,10], Takayuki Kato [7], Hiroshi Nagase [5], Ayori Mitsutake [4], Tsuyoshi Saitoh [5,11], Kota Katayama [2,12], Asuka Inoue [3,6], Hideki Kandori [2,12], Takuya Kobayashi [1,13] ✉ & Ryoji Suno [1] ✉

The κ-opioid receptor (KOR) is a member of the G protein-coupled receptor (GPCR) family, modulating cellular responses through transducers such as G proteins and β-arrestins. G-protein-biased KOR agonists aim to retain analgesic and antipruritic actions while limiting aversion and sedation. Aiming to inform G-biased KOR agonist design, we analyze signaling-relevant residues from structural and dynamic views. Here we show, using multiple complementary methods, shared residues that determine β-arrestin recruitment by nalfurafine and U-50,488H. Cryo-electron microscopy structures of the KOR-$G_i$ signaling complexes identify the ligand binding mode in the activated state. Vibrational spectroscopy reveals ligand-induced conformational changes. Cell-based mutant experiments pinpoint four amino acids (K227[5.40], C286[6.47], H291[6.52], and Y312[7.34]; Ballesteros–Weinstein numbering is shown in superscript) that play crucial roles in β-arrestin recruitment. Furthermore, MD simulations revealed that the four mutants tend to adopt conformations with reduced β-arrestin recruitment activity. Our research findings provide a foundation for enhancing KOR-mediated therapeutic effects while minimizing unwanted side effects by targeting specific residues within the KOR ligand-binding pocket, including K227[5.40] and Y312[7.34], which have previously been implicated in biased signaling.

G protein-coupled receptors (GPCRs) regulate intracellular processes through interactions with signal transducers like G proteins and β-arrestins. This process is triggered by the binding of extracellular agonists or, in the case of rhodopsin, by the photoisomerization of retinal[1]. Opioid receptors, members of the GPCR superfamily, are categorized into four subclasses: μ, δ, κ, and nociceptin receptors (MOR, DOR, KOR, and NOP receptors, respectively). These receptors

are recognized for their role in mediating the analgesic properties of opioid molecules[2].

Upon agonist binding, opioid receptors activate the $G_{i/o}$ subtype of G protein family, initiating a cascade of intracellular signaling pathways. This process attenuates the excitation of pain-sensing neurons and induces analgesia[3]. Studies with striatal neurons from β-arrestin knockout mice have revealed the involvement of the β-arrestin

pathway in mediating adverse effects, including drug aversion and sedation[4–6]. As a result, extensive research has been dedicated to developing G-protein-biased agonists that maintain maximal agonist activity while minimizing β-arrestin binding. The goal is to create analgesics that reduce side effects such as respiratory depression, drug dependence, sedation, and convulsions or catalepsy[7,8]. MOR agonists like morphine and fentanyl exhibit high analgesic efficacy but are linked with side effects such as respiratory depression and drug dependence. MOR-selective G-agonists, including TRV130, PZM21, and SR-17018, were developed to reduce these side effects[2]. Despite these innovations, fully suppressing adverse effects remains elusive. Agonists for KOR and DOR also lead to side effects like sedation and convulsions/catalepsy, respectively[9]. Several G-protein-biased ligands for DOR, including derivatives of SNC80 (e.g., ARM390), KNT-127, and those based on the morphinan backbone (SB-235863), have been developed to mitigate side effects[9,10]. Particularly, KNT-127 reduces the side effects of convulsions but is not approved for analgesia[11]. KOR agonists are crucial for pain relief, especially in the treatment of peripheral neuropathic pain, such as pruritus. However, their use is accompanied by side effects, including drug aversion, depression, sedation, and neuroendocrine disturbances[12]. Compounds like 6'-GNTI[13], triazole 1.1 and its derivatives[14], and salvinorin A derivatives[15] have been developed as G-protein-biased agonists[9].

Despite the development of biased agonists, the mechanisms underlying the selective signaling of GPCRs signaling induced by these compounds remain largely unexplored[2]. To comprehend the mechanism of action of drugs, the structures of opioid receptors bound to drugs with various properties have been determined. Further detailed analyses, including molecular dynamics (MD) simulations and pharmacological studies, reveal biases in ligand binding modes and mechanisms of action[16].

With respect to the structural biology of KOR, Daibani et al. recently determined the X-ray crystal structure of KOR bound to nalfurafine and a G protein mimicking nanobody (Nb39)[17]. They also modeled the ligand binding modes using the structures of KOR bound with the balanced agonist U-50,488H and the β-arrestin-biased agonist WMS-X600, followed by MD simulations of these complexes. MD simulations and pharmacological analysis of KORs bound with β-arrestin-biased and balanced agonists have demonstrated that the orientation of the side chain of Q115[2.60] and the distance between K227[5.40] and E297[6.58] differ for each ligand-bound state, thereby affecting signal selectivity. In addition, as part of their structural biological analysis, Daibani et al. also compared the active structures of nanobody-KOR complexes bound to nalfurafine and MP1104[17,18]. However, to date, no research has been conducted that compares the active structures of G protein-bound KOR to explore the biased signaling mechanism of the receptor. Additionally, experimental observations of agonist-dependent dynamic conformational changes in KORs prior to the binding of signaling molecules have not been made, and the information on selective signaling mechanisms is still developing.

In this study, we present cryo-EM structures of the KOR-$G_i$ signaling complexes bound with U-50,488H and nalfurafine. Furthermore, we analyze ligand-dependent dynamic changes in the amino acid side chains of KOR using attenuated total reflection-Fourier transform infrared spectroscopy (ATR-FTIR), a type of vibrational spectroscopy. Based on the structural information and the spectroscopy dynamics data, we conducted a mutagenesis study and identified amino acid residues (K227[5.40], C286[6.47], H291[6.52], Y312[7.34]) that play crucial roles in signaling via β-arrestin. We employed a variety of methods to show that four common amino acids in KOR contribute to signal selectivity in two agonists with different scaffolds. Additionally, MD simulations implied that the four mutants tend to adopt structures that reduce β-arrestin-recruitment activity.

## Results

### Comparison of the ligand-binding modes of nalfurafine- or U-50,488H-bound KOR-$G_i$ complexes

The cryo-EM structures of nalfurafine- or U-50,488H-bound human KOR-$G_i$ signaling complexes were determined at 2.76 Å and 2.9 Å resolution from 858,423 and 1,225,096 particles, respectively (Supplementary Figs. 1a–d, 2a–d, and 3a–d). This allowed us to precisely identify and assign the transmembrane (TM) domain of KOR, the $G_i$ protein heterotrimer, the antibody fragment, and ligands in the cryo-EM map (Fig. 1a, b and Supplementary Fig. 4a, b). Overlaying the receptor regions of the nalfurafine- and U-50,488H-bound KOR-$G_i$ complexes revealed a close alignment with a backbone root mean square deviation (RMSD) of 0.4 Å. Analysis of the cryo-EM data from the nalfurafine-bound KOR-$G_i$ complex yielded four different maps, the highest resolution of which was used for structure building and final refinement. When we constructed for all four maps and evaluated the differences, the receptor-only structures showed minimal validation, with RMSD values ranging from 0.19 to 0.25 Å. In contrast, the relative positioning of the G proteins in relation to the receptor varied, suggesting different states of G protein binding to the KOR (Supplementary Fig. 2e).

To elucidate the differences in the binding modes of U-50,488H and nalfurafine to KOR, we compared the structures of the two KOR-$G_i$ complexes determined in this study with the previously reported MP1104 (a non-selective balanced agonist of opioid receptors)-bound state[18]. The major difference observed in the ligand binding mode was the interaction of nalfurafine with TM5, particularly K227[5.40]. In contrast, U-50,488H and MP1104 exhibited no interaction with K227[5.40] (Figs. 1c, d and 2a).

Structural determination of the KOR-$G_i$ signaling complexes in nalfurafine- and U-50,488H-bound forms unveiled the details of their respective agonist binding modes (Fig. 1d). Key residues, including K227[5.40], W287[6.48], and I294[6.55] located in TM 5 and 6, exhibited exclusive van der Waals interactions with nalfurafine, while no interaction with U-50,488H was observed. In TM2, both agonists formed van der Waals interactions with the side chains of V108[2.53]; the side chain of Q115[2.60] formed a van der Waals interactions with U-50,488H and a hydrogen bond with nalfurafine.

In TM3, the side chain of V134[3.28] formed a van der Waals interaction only with nalfurafine. The side chain of D138[3.32] forms van der Waals interactions with U-50,488H and forms hydrogen and ionic bonds with nalfurafine. Similarly, the side chain of Y139[3.33] interacts with U-50,488H via van der Waals forces but forms hydrogen bonds with nalfurafine. In TM7, both Y312[7.34] and Y320[7.42] interact with the agonists through van der Waals interactions.

To validate the functional importance of agonist-interacting residues, we conducted a mutagenesis study using NanoBiT-based G protein-dissociation assay and β-arrestin-recruitment assay. Using the NanoBiT assay to assess the bias of the two ligands, we found no detectable bias in either (Fig. 1e–g). Mutants were chosen based on amino acids located within less than 4 Å from the ligand, as depicted in Fig. 1d. The expression levels of mutants were confirmed by flow cytometry (Supplementary Fig. 5). Overall, the G protein activity ($E_{max}$) of each mutant remained largely unaffected by any ligand (Supplementary Figs. 6 and 7, Supplementary Table 1 and Supplementary Data 1). However, the K227[5.40]A and Y312[7.34]A/F mutants exhibited a significant reduction in $E_{max}$ for β-arrestin-recruitment activity with both ligands (Supplementary Figs. 8 and 9, Supplementary Table 1 and Supplementary Data 1). In addition to the $G_{i1}$ dissociation reaction shown in Supplementary Fig. 8, we explored differences in signal transduction among the G protein subtypes of the mutants. The decrease in $pEC_{50}$ values varied depending on the agonist and specific mutation, but the overall balance between the $G_{i/o}$ protein subtypes was consistent across the mutants. Notably, $G_z$ signaling showed

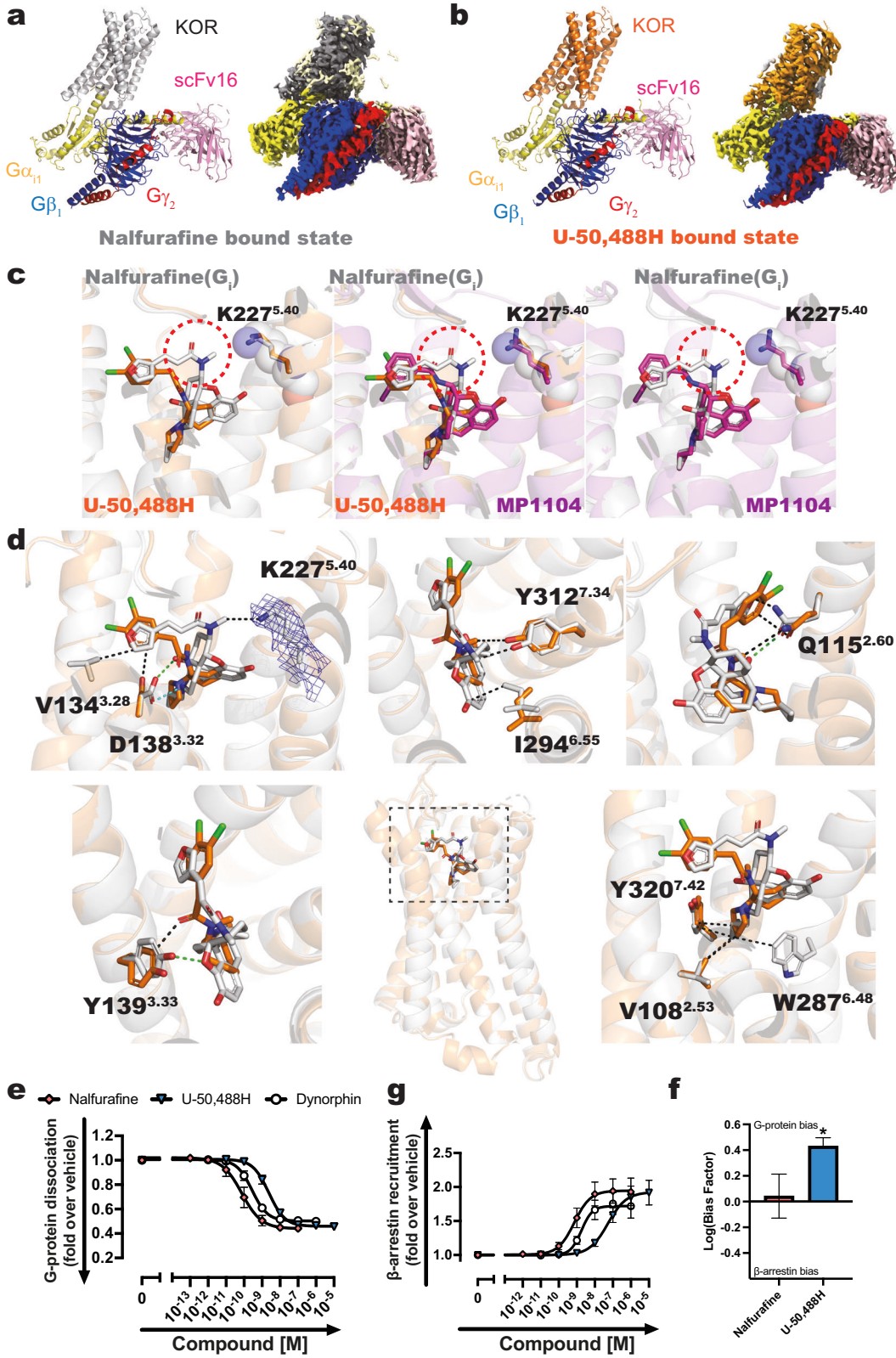

variations in $E_{max}$ values for the mutants, especially with U-50,488H (Supplementary Fig. 10).

## Identification of crucial amino acid residues for β-arrestin recruitment

In addition to signal assays with agonists, structural comparisons of the KOR-$G_i$ signaling complexes bound to nalfurafine and U-50,488H were conducted to investigate two key amino acids influencing β-arrestin recruitment, initially focusing on K227[5.40]. Comparative structural analysis of the two KOR-$G_i$ signaling complexes revealed direct interaction of the side chain of K227[5.40] with nalfurafine but not with U-50,488H. In the U-50,488H-bound KOR structure, a weak ionic bond was observed between the side chains of K227[5.40] and E297[6.58], absent in the nalfurafine-bound KOR structure (Fig. 2a).

**Fig. 1 | Cryo-EM structures of the agonist-bound KOR-$G_i$ signaling complexes.** Comparison of the structural model with the cryo-EM density of the entire complex. Structures and cryo-EM maps of the KOR-$G_i$ complex in nalfurafine- (**a**) and U-50,488H-bound (**b**) states. Nalfurafine-bound KOR is gray, U-50,488H-bound KOR is displayed orange, $G_i$ is yellow, $G_\beta$ is blue, $G_\gamma$ is red, and scFv16 is pink. **c** Differences in ligand binding mode of nalfurafine (gray), U-50,488 (orange), and MP1104 (magenta). The red dotted circles indicate characteristic regions of nalfurafine's structure compared to other agonists. **d** Amino acid residues of KOR that interact with nalfurafine or U-50,488H. Van der Waals interactions are represented by black, green, and cyan dotted lines for van der Waals forces, hydrogen bonds, and ionic bonds, respectively. Amino acids that are within 4 Å of the ligand are displayed as sticks. The cryo-EM map is displayed as blue mesh. **e, f** G-protein- and β-arrestin-mediated signals were examined with the NanoBiT-G dissociation (**e**) and NanoBiT-β-arrestin recruitment assay (**f**), respectively. Dynorphin, an endogenous agonist of KOR, was used as a reference ligand. In the concentration-response curves, symbols represent the mean, and error bars indicate the SEM from three independent experiments, each conducted in duplicate. **g** Ligand bias calculated from the data shown in (**a, b**). The $\Delta\Delta Log(E_{max}/EC_{50})$ values were used to evaluate ligand bias, as described previously (Kolb et al., 2024, https://doi.org/10.1111/bph.15811). Specifically, the relative activity of nalfurafine or U-50,488H to dynorphin was calculated in each signaling assay, and these relative activities were then compared between assays. Statistical significance was calculated by one sample $t$-test (ns, $p > 0.05$; *, $p < 0.05$; **, $p < 0.01$; ***, $p < 0.001$).

In the NanoBiT assays, the K227[5.40]A mutation did not affect the $E_{max}$ of G protein coupling but significantly reduced the $E_{max}$ of β-arrestin-recruitment activity in the presence of both agonists. Conversely, K227[5.40]A exhibited largely unchanged G protein coupling with nalfurafine, albeit with reduced $pEC_{50}$ in the presence of U-50,488H (Fig. 2b, Supplementary Figs. 6–9, Supplementary Table 1 and Supplementary Data 1). Although there is no direct interaction between K227[5.40] and U-50,488H, the K227[5.40]A mutation led to a significant decrease in signaling activity (Fig. 2b). Given the potential role of the K227[5.40]-E297[6.58] interaction, we created an E297[6.58]A mutant to assess its impact on signaling activity. The E297[6.58]A mutant exhibited activity nearly identical to the wild-type (Supplementary Fig. 11).

Next, we examined the Y312[7.34]A and Y312[7.34]F mutants, which demonstrated a significant reduction in the $E_{max}$ of β-arrestin-recruitment activity when treated with nalfurafine and U-50,488H (Fig. 3a, Supplementary Figs. 6–9, Supplementary Table 1 and Supplementary Data 1). Specifically, the Y312[7.34]A mutant exhibited a markedly decreased $pEC_{50}$ for G protein-coupling activity and a substantially lower $E_{max}$ for β-arrestin-recruitment activity with both ligands. Although the G protein-coupling activity of the Y312[7.34]F mutant with nalfurafine resembled that of the wild-type, its $pEC_{50}$ with U-50,488H decreased. These findings imply that the hydroxyl group of the Y312[7.34] side chain plays a crucial role in nalfurafine-mediated β-arrestin-recruitment activity.

The distances from the hydroxyl group on the side chain of Y312[7.34] to U-50,488H, nalfurafine, and MP1104 were measured at 3.9, 3.3, and 3.6 Å, respectively (Fig. 3b). The Y312[7.34] mutants exhibited different distances from the ligand, which is thought to weaken the van der Waals interaction. To determine if the changes in signaling activity observed in the Y312[7.34] mutants were due to differences in ligand binding affinity, we conducted ligand binding experiments with the Y312[7.34]A/F mutants and the wild-type. Using [$^3$H] U-69,593 as the radioactive ligand, the $K_d$ values for the wild-type, Y312[7.34]A, and Y312[7.34]F were calculated as 7.79, 12.1, and 6.29 nM, respectively, indicating that the Y312[7.34]A mutation slightly reduced affinity (Supplementary Fig. 12a). We then performed competitive binding experiments with nalfurafine and U-50,488H to determine the $K_i$ values. The results consistently showed that nalfurafine had a higher binding affinity than U-50,488H across all mutants. These binding experiment results suggest that the Y312[7.34]A/F mutations do not significantly impair binding affinity, indicating that Y312[7.34] is not essential for ligand binding (Supplementary Fig. 12b). Overall, we concluded that while Y312[7.34] is not critical for agonist binding, the interaction between the side chain of Y312[7.34] and the agonist play a key role in β-arrestin recruitment activity.

Based on MD simulations, Daibani et al. identified Q115[2.60] as an amino acid involved in KOR signaling[17]. To understand the role of Q115[2.60], we compared the structures of KOR-$G_i$ signaling complexes bound to nalfurafine or U-50,488H (Supplementary Fig. 13). In the KOR structure bound to U-50,488H, the side chain of Q115[2.60] engaged in a weak van der Waals interaction with the ligand and formed a hydrogen bond with the side chain of Y320[7.42]. Conversely, nalfurafine interacted with the side chain of Q115[2.60] at two locations, establishing both a van der Waals interaction and a hydrogen bond. This suggests that nalfurafine forms a stronger interaction with the side chain of Q115[2.60] than U-50,488H. Previous MD simulations have indicated that when KOR binds to WMS-X600, a β-arrestin-biased agonist, the side chain of Q115[2.60] orients towards both Y320[7.42] and Y66[1.39] [17]. These observations suggest that differences in agonist binding with Q115[2.60] result in different side chain orientations of Q115[2.60], which affect β-arrestin-recruitment activity (Supplementary Fig. 13).

## Quantifying agonist-dependent conformational changes in KOR using infrared spectroscopy

The active state of KOR bound to G protein, as determined by cryo-EM, reveals the agonist binding mode, although the structure represents the conformation after G protein binding. The binding of an agonist to KOR may create an environment conducive to the binding of each signaling transduction factor. However, to gain detailed structural insights, it is essential to detect the structural changes that occur prior to G protein and β-arrestin binding. Therefore, we next investigated the conformational changes that take place upon agonist binding to KORs, before the recruitment of signal transducers, using ATR-FTIR spectroscopy[19–21] (Fig. 4a, b).

When nalfurafine and U-50,488H were introduced to the inactive state of KOR bound to the antagonist naltrexone, distinct spectral changes were detected in the 1666–1650 cm$^{-1}$ region of the amide-I band. These changes were attributed to conformational shifts in the receptor's α-helix. In addition to alterations in the receptor's secondary structure, differences were also observed in the vibrational frequency regions, including the C-N stretching vibration of histidine (1200–1100 cm$^{-1}$)[22] and the S-H stretching vibration of cysteine (2600–2500 cm$^{-1}$)[23]. The band around 2550 cm$^{-1}$, characteristic of the S-H stretching vibration of cysteine, exhibited a band-shift for both ligand-bound forms, indicating a difference in the hydrogen-bonding environment of cysteine (refer to Fig. 4a). In the inactive state bound to the antagonist naltrexone (negative side), a consistent band was observed at 2550 cm$^{-1}$. In contrast, in the active states induced by the agonists nalfurafine and U-50,488H (positive side), the band shifted up to 2570 and 2565 cm$^{-1}$, respectively. These upshifts suggest that agonist binding weakens or disrupts the hydrogen bond involving cysteine. A similar trend was observed in the protein backbone. As shown in Fig. 4b, ligand exchange from naltrexone to nalfurafine or U-50,488H resulted in an upshift of the amide-I band from 1651 cm$^{-1}$ to 1666 and 1663 cm$^{-1}$, respectively. This spectral shift is consistent with a perturbation of α-helical hydrogen bonding networks, particularly involving the backbone C = O groups. Such upshifts typically reflect a weakening of intrahelical hydrogen bonds, possibly due to subtle local perturbation or changes in helical packing induced by agonist binding. Conversely, in Fig. 4b, the negative bands of the naltrexone-bound form are identical to the red and blue lines, yet a 3 cm$^{-1}$ variance is observed between the nalfurafine-bound form (red line: 1666 cm$^{-1}$) and the U-50,488H-bound form (blue line: 1663 cm$^{-1}$). This disparity suggests that nalfurafine binding induces more local α-helical

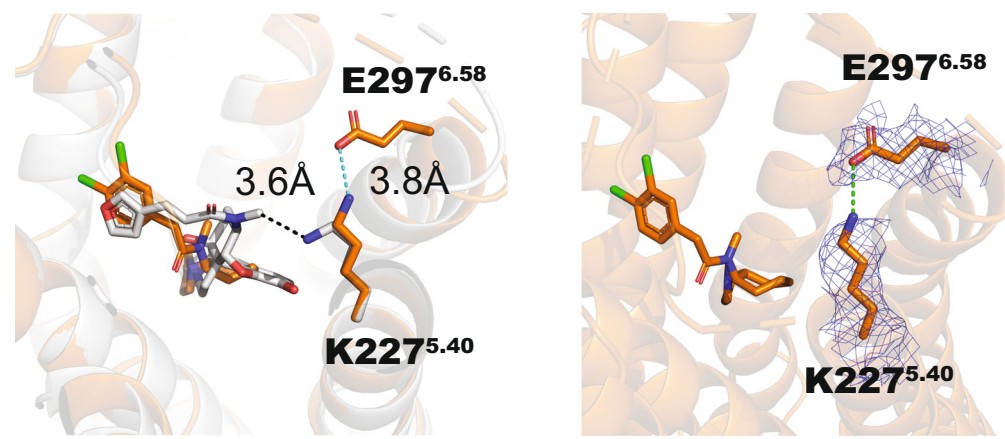

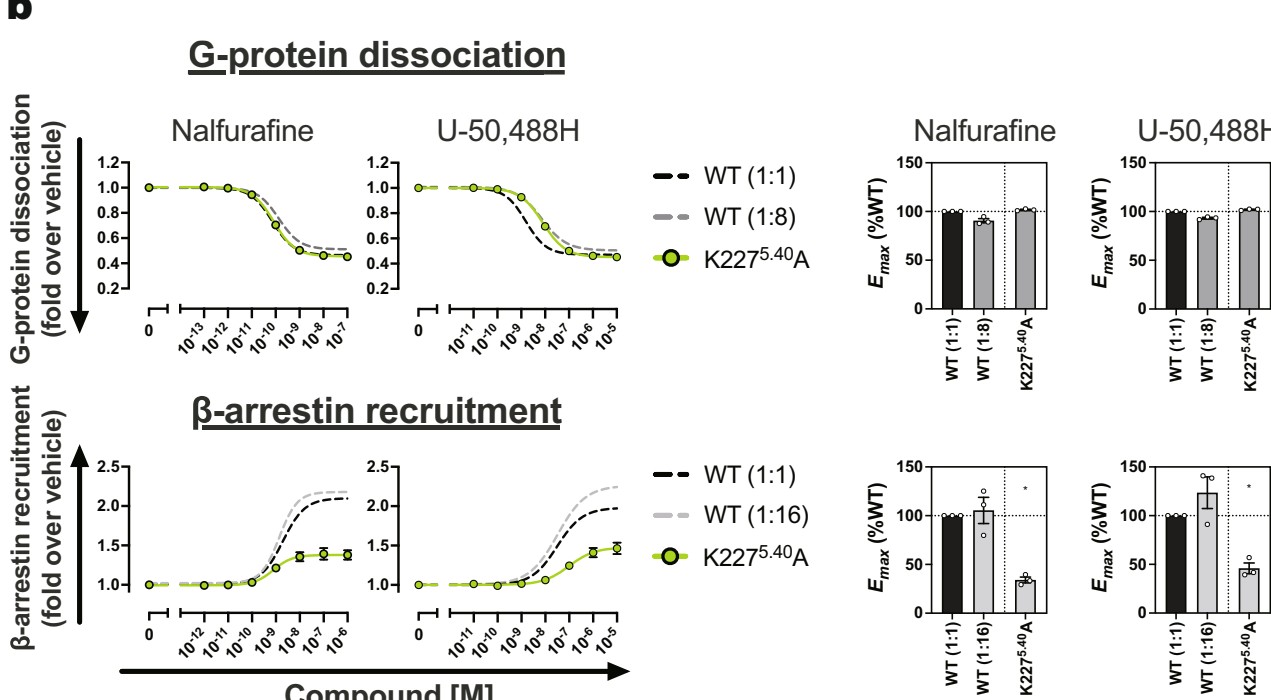

**Fig. 2 | Contribution of K227^5.40 to the signal selectivity of KOR.** Binding mode of the side chain of K227^5.40 to the agonists (**a**). Van der Waals interactions are depicted in black dotted lines, while hydrogen bonds and ionic bonds are represented by green dotted lines. Amino acids that are within 4 Å of the ligand are displayed as sticks. Nalfurafine-bound KORs are represented in gray, U-50,488H-bound KORs are orange. The cryo-EM maps are displayed as blue mesh. Concentration-response curves of the NanoBiT-G protein dissociation assay and the NanoBiT-β-arrestin recruitment assay of the K227^5.40 (**b**) mutant. Dashed lines in the mutant panels represent the wild-type KOR (1:1) response. Data are presented as mean values ± SEM (*n* = 3). Note that in numerous data points, the error bars are smaller than the size of the symbols, making them not visible. Data from the wild-type with reduced surface expression serve as a reference for comparing mutants that show similar reductions in expression levels (Supplementary Fig. 5). Wild-type (WT) (1:2), (1:4), etc., represents plasmid amounts reduced to 1/2, 1/4, and so on, relative to the standard level for transfection.

perturbation. Furthermore, the positive 1130 cm$^{-1}$ band likely reflects the characteristic C-N stretching vibration of histidine imidazole[22] and is amplified upon binding of nalfurafine.

To elucidate the role of specific amino acid residues (cysteine and histidine) identified by FTIR spectroscopy in ligand-induced conformational changes and their relevance to KOR activation, we conducted functional assays using a series of point mutants. Structural information of KOR reveals three cysteines (C229^5.41, C286^6.47, and C315^7.37) that do not directly interact with the agonist. These cysteine residues were substituted with alanine, and their impact on signaling activity was assessed. The C229^5.41A mutation exhibited no discernible change in G protein binding and β-arrestin recruitment. However, the C286^6.47A mutant displayed a considerable decrease in $E_{max}$ of β-arrestin-recruitment activity for both ligands, while the C315^7.37A mutant exhibited a decrease in G protein pEC$_{50}$ and a slight decrease in β-arrestin-recruitment activity only in the presence of U-50,488H (Fig. 4c, Supplementary Figs. 6–9, Supplementary Table 1 and Supplementary Data 1). Specifically focusing on C286^6.47A, which

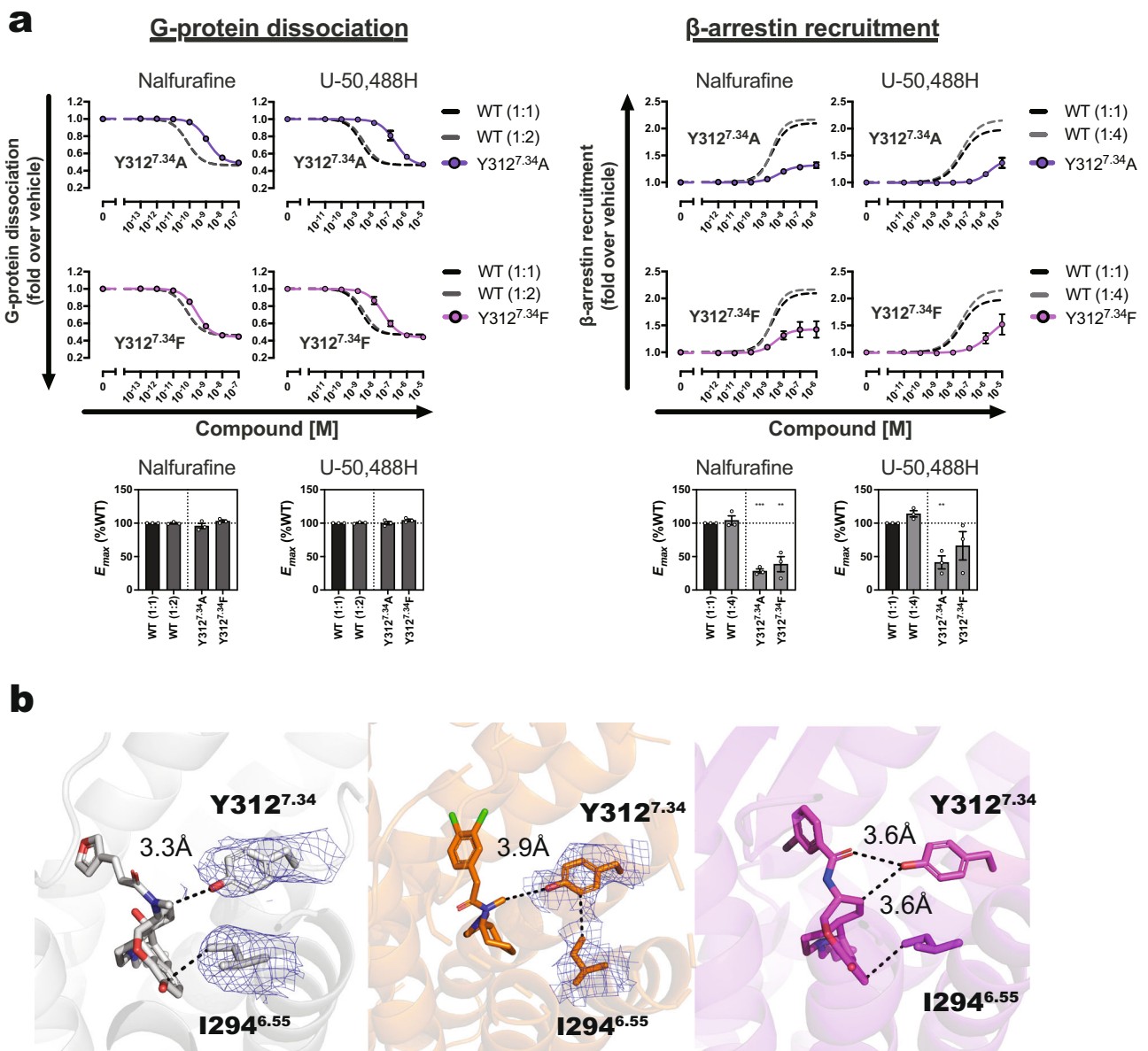

**Fig. 3 | Contribution of Y312[7.34] to the signal selectivity of KOR.** Concentration-response curves of the NanoBiT-G protein dissociation assay and the NanoBiT-β-arrestin recruitment assay of Y312[7.34] (**a**) mutant. Dashed lines in the mutant panels represent the wild-type KOR (1:1) response. Data are presented as mean values ± SEM (*n* = 3). Note that in numerous data points, the error bars are smaller than the size of the symbols, making them not visible. Data from the wild-type with reduced surface expression serve as a baseline for comparison with mutants that exhibit similar reductions in expression levels (Supplementary Fig. 5). Wild-type (WT) (1:2), (1:4), etc., represents plasmid amounts reduced to 1/2, 1/4, and so on, relative to the standard level for transfection. Binding mode of the side chain of Y312[7.34] to the agonists (**b**). Nalfurafine-bound KOR is represented in gray, U-50,488H-bound KOR is orange, and MP-1104-bound KOR is magenta. The cryo-EM maps are displayed as blue mesh.

demonstrated a significant change in signaling activity, no observable interactions were noted between the side chains of C286[6.47] and T321[7.43] in the inactive state, suggesting that these interactions are promoted in an agonist binding-dependent manner (Fig. 4d and Supplementary Fig. 14).

The mutation C286[6.47]A resulted in a greater distance from the side chain of T321[7.43] compared to the wild-type, indicating a weakened interaction. This observation, coupled with the mutagenesis analysis, suggests that the interaction between the side chains of C286[6.47] and T321[7.43] is essential for β-arrestin-recruitment activity. Specifically, the G protein-coupling activity of C286[6.47]A was nearly identical to that of the wild-type upon the addition of nalfurafine, implying that this interaction does not impact G protein dissociation. Regarding C315[7.37], its side chain

interacts with the oxygen of the main chain carbonyl group of P289[6.50] (Supplementary Fig. 15). In the presence of U-50,488H, both the G protein dissociation and β-arrestin-recruitment activities of C315[7.37]A were slightly impaired (Supplementary Figs. 6–9, Supplementary Table 1, and Supplementary Data 1). These findings suggest that the interaction of C315[7.37] side chain with the oxygen of the P289[6.50] carbonyl group is crucial for forming a pocket suitable for U-50,488H binding. The FTIR ligand-induced difference spectra revealed a weakening of the hydrogen bonding of cysteine upon agonist binding (Fig. 4a). While our FTIR measurements captured conformational changes upon ligand exchange from naltrexone, the structural comparison shown in Fig. 4d was based on the JDTic-bound inactive conformation. Thus, the structural transitions may not be entirely equivalent. Nevertheless, the

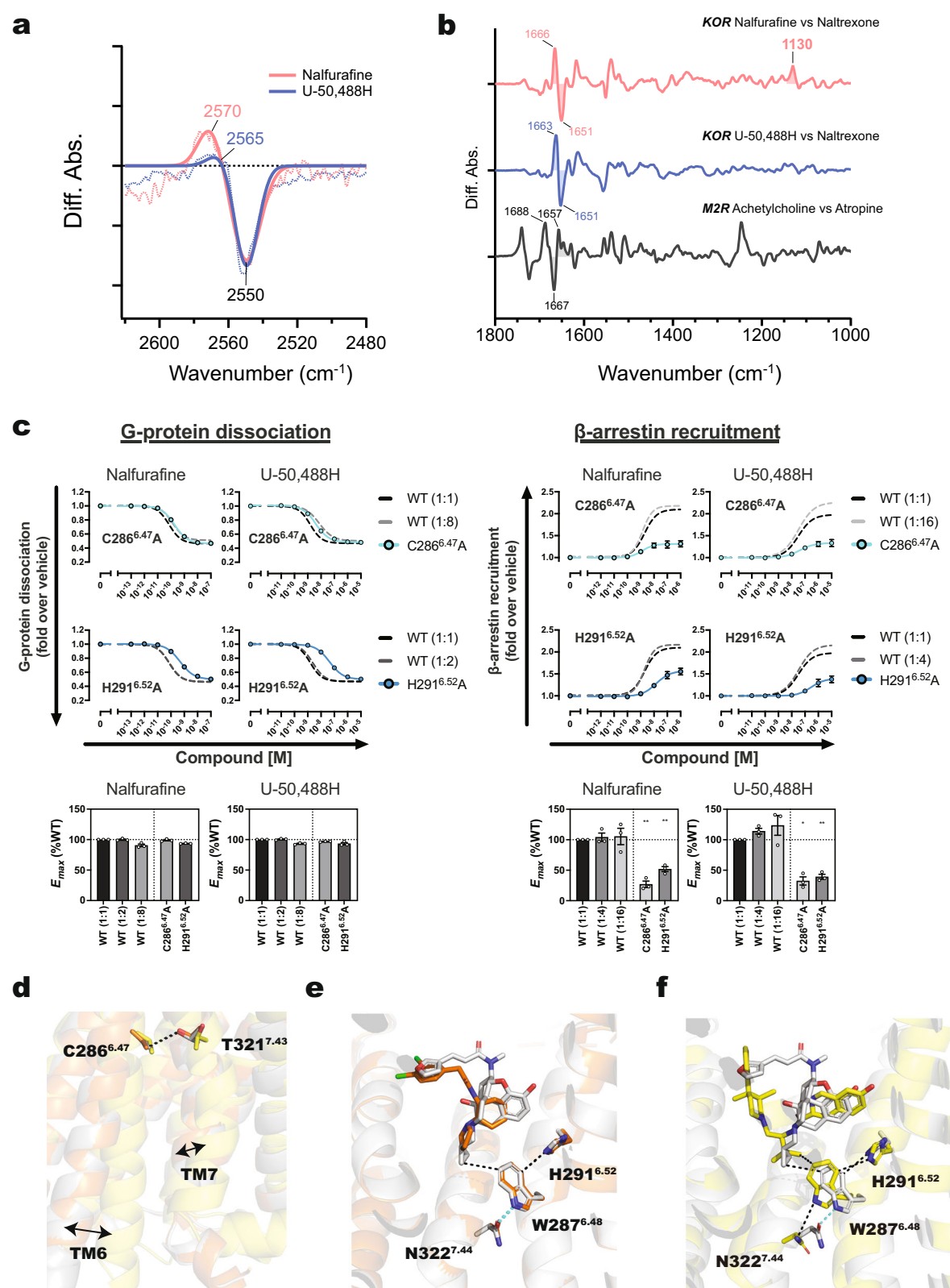

combined evidence from structural modeling, spectroscopy, and functional assays strongly implicates C286[6.47] and C315[7.37] as key residues modulating biased signaling in KOR.

Next, histidine residues in KOR associated with signaling activity were investigated through mutagenesis. Two histidine residues of KOR, H162[ICL2] and H291[6.52], were selected for structural analysis. KOR-$G_i$ signaling structures show that the side chain of H162[ICL2] remained

exposed in solution without interaction with any amino acid side chain or ligand. Conversely, structural data from the KOR-$G_i$ complex in the nalfurafine-bound state confirmed an interaction between the side chains of H291[6.52] and W287[6.48] (Fig. 4e, f).

The mutagenesis assay revealed that the H291[6.52]A mutant reduced both G protein dissociation and β-arrestin-recruitment activities. Notably, the decrease in $E_{max}$ of β-arrestin-recruitment activity

**Fig. 4 | Ligand binding-dependent conformational changes detected by ATR-FTIR and their effect on signal selectivity.** FTIR spectroscopy was employed to detect conformational changes in the KOR upon agonist binding. Signals from nalfurafine and U-50,488H were observed as they interacted with the KOR in the presence of the antagonist naltrexone. ATR-FTIR spectra illustrating the conformational changes in cysteine (**a**) and histidine (**b**). **c** Concentration-response curves of the NanoBiT-G protein dissociation assay and the NanoBiT-β-arrestin recruitment assay of C286[6.47]A and H291[6.52]A mutants. Dashed lines in the mutant panels represent the wild-type (WT) KOR (1:1) response. Data are presented as mean values ± SEM ($n = 3$). Note that in numerous data points, the error bars are smaller than the size of the symbols, making them not visible. **d** Interaction between C286[6.47] and T321[7.43] and conformational changes in TM6 during agonists and antagonist binding. Arrows indicate the extent of conformational change in TM6 when comparing inactive and active forms. **e** An interaction network is formed between agonists, H291[6.52] and N322[7.44], which links with a toggle switch (W287[6.48]) that is common to GPCRs and important for their activation. **f** Interaction of H291[6.52] and W287[6.48] in agonist- or antagonist-bound states. Nalfurafine-bound KORs are represented in gray, U-50,488H-bound KORs are in orange, and inverse agonist JDTic-bound KORs are displayed in yellow. Data from the wild-type, which exhibits reduced surface expression, serve as a reference for comparing mutants that show similar reductions in expression levels. Wild-type (WT) (1:2), (1:4), etc., represents plasmid amounts reduced to 1/2, 1/4, and so on, relative to the standard level for transfection.

highlights the importance of side chain interactions between W287[6.48] and H291[6.52] in the β-arrestin response (Fig. 4c, Supplementary Figs. 6–9, Supplementary Table 1, and Supplementary Data 1). A noticeable rotation of the side chain of H291[6.52] was observed in the agonist-bound state of KOR in comparison to the inverse agonist JDTic-bound state (Fig. 4f), which was identified as a signal in the infrared spectroscopy.

In the nalfurafine-bound state, when the interaction distance is defined as within 4 Å, W287[6.48] directly interacts with the ligand, while H291[6.52] shows no direct interaction with either ligand. This suggests that H291[6.52] is crucial for stabilizing the orientation of the W287[6.48] side chain, indirectly influencing ligand binding and signaling activity (Fig. 4e, f). Despite the distance between U-50,488H and W287[6.48] being more than 4 Å, both W287[6.48]A and H291[6.52]A mutants significantly reduced β-arrestin-recruitment activity like the addition of nalfurafine, indicating a molecular mechanism for β-arrestin recruitment similar to nalfurafine.

Furthermore, H291[6.52] in TM6 interacts with W287[6.48], forming an interaction network with N322[7.44] of TM7. These interactions underscore the significant role played by the interplay between TM6 and TM7 in the β-arrestin-recruitment activity of KOR (Fig. 4e, f, Supplementary Figs. 6–9, Supplementary Table 1 and Supplementary Data 1). Notably, W287[6.48], a well-characterized toggle switch involved in GPCR activation, is in close spatial proximity to the newly identified residues C286[6.47], C315[7.37], and H291[6.52]. Ligand-induced conformational changes originating from the toggle switch may cause a localized kink in TM6, which likely corresponds to the amide-I band upshift observed in our FTIR spectra.

## Conformational landscape changes in K227[5.40]A, Y312[7.34]A, C286[6.47]A, and H291[6.52]A mutants

To investigate the structural dynamics associated with β-arrestin-recruitment activity, we conducted three independent one-microsecond MD simulations for each of the wild-type (WT) and four mutants (K227[5.40]A, Y312[7.34]A, C286[6.47]A, and H291[6.52]A), all complexed with nalfurafine in the absence of G protein, totaling 15 microseconds of simulation time (Supplementary Table 2). The time series of RMSDs are summarized in Supplementary Figs. 16–18.

First, interaction fingerprint analysis was performed to identify the amino acid residues binding to the ligand within 6 Å[24]. The most frequently interacting amino acid residues in the WT receptor are summarized in Supplementary Fig. 19. For those amino acid residues that exhibited interactions with the ligand, we compared the changes in interaction frequencies across the four mutants. In Supplementary Fig. 19, positive values indicate a reduction in interaction due to the mutations, while negative values indicate an increase. Notably, interactions between the ligand and residues H291[6.52], W124[ECL1], and Q115[2.60] were diminished in the mutants relative to the WT. On the other hand, interactions with Y320[7.42] and S211[45.51] were enhanced. The observed changes in Y320[7.42] and Q115[2.60] are consistent with previous results[17] and our discussion, supporting the hypothesis that Q115[2.60] may play a role in β-arrestin-recruitment activity.

Next, we focus on the interaction site of G protein and β-arrestin, especially the state of R156[3.50]. From these extensive simulations, we observed a decrease in the populations of both the alternative conformations and the conformations where R156[3.50] faces the intracellular side in all mutants compared to the WT (Fig. 5 and Supplementary Fig. 20). The amino acid residues in the intra-cellular regions of GPCRs have been extensively studied to elucidate the mechanism of selectivity between G protein and β-arrestin. The canonical active and alternative conformations have also been well-characterized and actively discussed[25]. The canonical active conformation represents the typical active state of GPCRs that facilitates both G protein signaling and β-arrestin recruitment. In contrast, the alternative conformation is considered an intermediate state that may preferentially favor β-arrestin recruitment over G protein activation, although it has not yet been observed in experimental structures[25]. The alternative conformation, in which R156[3.50] and Y330[7.53] face the intracellular side, represents a receptor state that might preferentially promote β-arrestin recruitment. This conformation is believed to result from structural rearrangements that facilitate β-arrestin binding while hindering G protein coupling. Specifically, the rotamer conformation of R156[3.50] differs significantly between the KOR-G protein complex and the KOR-β-arrestin complex, leading to differences in signal selectivity. When a GPCR binds to a G protein, R156[3.50] primarily faces the extracellular side[26], while it faces intracellularly when interacting with β-arrestin[27]. By quantitatively analyzing the MD simulations for the WT and four mutants, using the interatomic distance between S153[3.47] and Y330[7.53] along with the dihedral angle of Y330[7.53], we identified four distinct conformational states on the intracellular side of the KOR (Fig. 5a–e). These states were further categorized based on the orientation of R156[3.50], specifically whether the side chain faced the intracellular side (downward). The percentages shown in Fig. 5a–e represent the proportion of structures within each state where R156[3.50] adopts this intracellular-facing orientation. Since states 1 and 4 exhibited little to no populations with the R156[3.50] facing downward, only states 2 and 3 were divided into two sub-states each: 2-A, 2-B, 3-A, and 3-B. As a result, six major states were defined: state 1, state 2-A, state 2-B, state 3-A, state 3-B, and state 4 (Fig. 5f). State 1 corresponds to the canonical active conformation, with both R156[3.50] and Y330[7.53] facing the extracellular side (Fig. 5g). State 2-A represents the conformation with R156[3.50] oriented intracellularly (Fig. 5h), while state 2-B resembles the canonical active conformation (Fig. 5i). State 3-A corresponds to the alternative conformation, where both R156[3.50] and Y330[7.53] face intracellularly (Fig. 5j), whereas state 3-B represents other conformations (Fig. 5k). State 4 is a unique conformation observed only in the MD simulations, characterized by the side chain of Y330[7.53] facing toward the intracellular side of TM1 (Fig. 5l).

Compared to the WT, the K227[5.40]A mutant tends to exhibit a reduction in the population of state 3-A, corresponding to the alternative conformation with R156[3.50] facing intracellularly (Fig. 5f). The Y312[7.34]A mutant showed a decrease in the population of state 3-

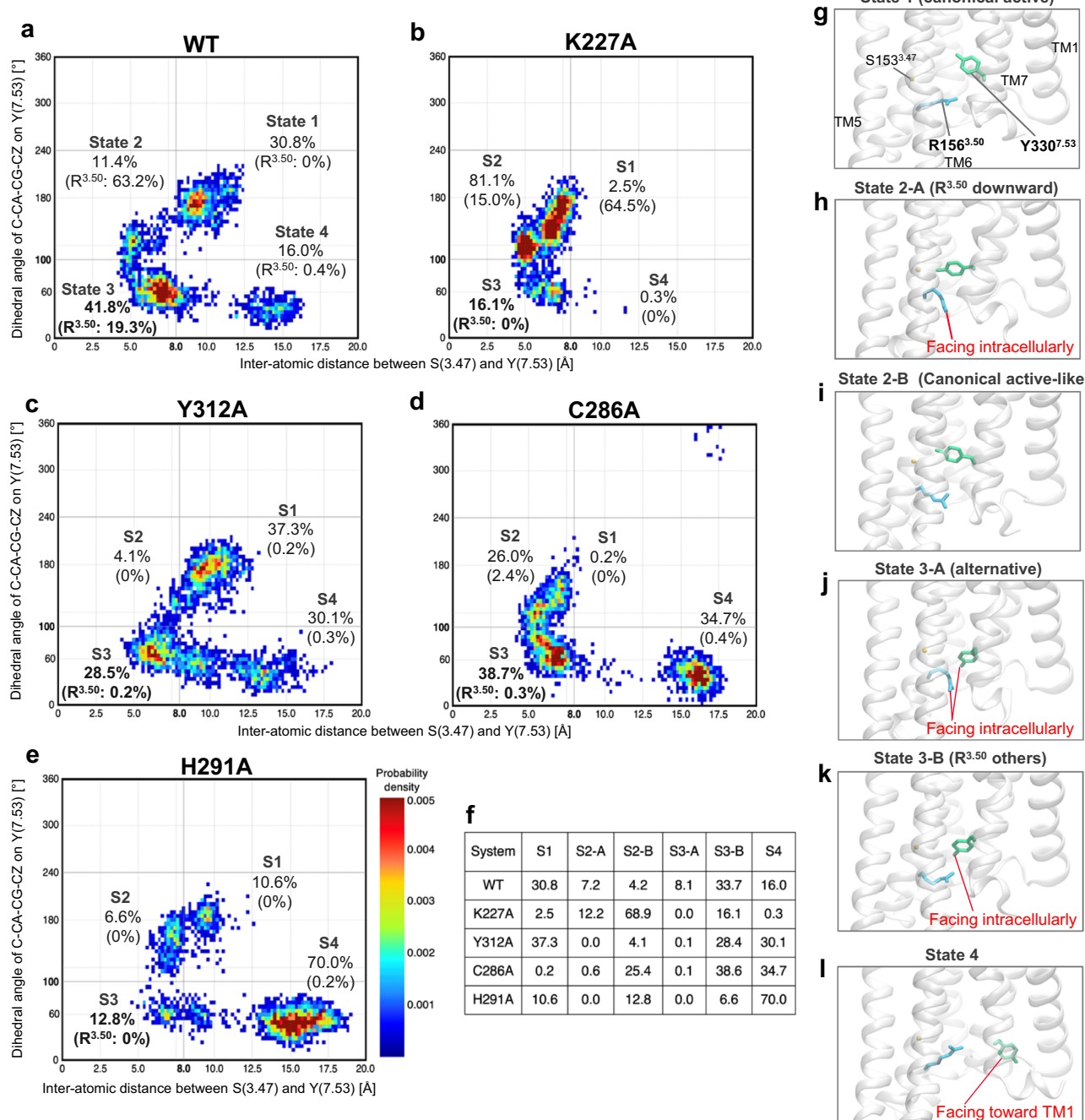

**Fig. 5 | Quantitative evaluation of intracellular conformations in the WT and K227[5.40]A, Y312[7.34]A, C286[6.47]A, and H291[6.52]A mutants using MD simulations.** The populations of both the alternative conformations and those in which R156[3.50] faces toward the intracellular side decreased significantly in all mutants compared to the WT. Heatmap of the intracellular conformations based on the interatomic distance between Cα of S153[3.47] and Cζ of Y330[7.53] and the dihedral angle of C–Cα–Cγ–Cζ of Y330[7.53] for the WT and the four mutants are shown in (**a**–**e**): **a** WT, **b** K227[5.40]A, **c** Y312[7.34]A, **d** C286[6.47]A, and **e** H291[6.52]A. Four major states were identified (states 1–4) and further subdivided based on the orientation of R156[3.50]: states 2 and 3 were split into two substates (2-A, 2-B, 3-A, and 3-B), resulting in six defined

conformational states. The percentages represent the populations of each state across the three independent one-microsecond MD simulations. The percentages labeled R[3.50] indicate the proportion of structures within each state in which R156[3.50] faces toward the intracellular side. The proportions of all six conformational states for each system are summarized in (**f**). **g**–**l** depict representative conformations of each state observed in the WT: **g** State 1 (S1), canonical active conformation; **h** S2-A, the conformation with R156[3.50] facing intracellularly; **i** S2-B, similar to the canonical active conformation; **j** S3-A, alternative conformation with both R156[3.50] and Y330[7.53] facing intracellularly; **k** S3-B, other conformations in state 3; **l** S4, a unique conformation characterized by Y330[7.53] facing toward the intracellular side of TM1.

A, with rare occurrences of R156[3.50] facing the intracellular side, suggesting a potential reduction in β-arrestin recruitment activity. In the C286[6.47]A mutant, the overall population of state 3-A decreased, and the canonical conformation (state 1) was markedly reduced. The H291[6.52]A mutant demonstrated decrease in states 1

and 3-A, accompanied by a relative increase in state 4. Overall, these findings indicate that mutations at positions K227[5.40], Y312[7.34], C286[6.47], and H291[6.52] alter the conformational landscape of the KOR, leading to reduced populations of states associated with β-arrestin recruitment activity.

### Structural comparison between the nanobody-bound KOR and the KOR-G$_i$ complex in nalfurafine-bound state

Recently, the structure of the nalfurafine-bound KOR was elucidated using X-ray crystallography in its nanobody (Nb39)-bound state, which stabilizes its active conformation[17]. In this arrangement, both the C-terminal regions of Nb39 and G$_i$ enter and bind to the intracellular pocket of the KOR.

A comparative structural analysis of the KOR complex in these nalfurafine-bound states highlighted minor conformational variations in amino acid side chains, along with three major differences. Firstly, in the Nb39-bound state, we observed a notable outward opening of the intracellular region of TM5 and TM6, as illustrated in Supplementary Fig. 21a. Although the C-terminal region of G$_i$ penetrates the KOR more deeply compared to Nb39, the loop region of Nb39 exhibits significant interaction with the KOR pocket, likely influencing the observed conformational changes in TM5 and TM6 (Supplementary Fig. 21b). Secondly, a noticeable difference was observed in the orientation of the side chains of residues Y140$^{3.34}$ and W183$^{4.50}$ (Supplementary Fig. 21c). In the nalfurafine-bound form of the KOR-G$_i$ complex, the side chain of W183$^{4.50}$ underwent a significant shift, resulting in the formation of a hydrogen bond with Y140$^{3.34}$. To evaluate the importance of the interaction between Y140$^{3.34}$ and W183$^{4.50}$ on G protein dissociation and β-arrestin recruitment, we examined the signaling activity of Y140$^{3.34}$F and W183$^{4.50}$A mutants. Surprisingly, these amino acid substitutions did not substantially affect G protein dissociation and β-arrestin recruitment. Furthermore, in the presence of U-50,488H, the conformations of Y140$^{3.34}$ and W183$^{4.50}$ closely resembled those observed in the nalfurafine-bound KOR-Nb39 structure (Supplementary Fig. 21c). Additionally, slight differences in the rotamers of amino acid sidechains involved in β-arrestin-recruitment activity were found in this study (Supplementary Fig. 21d).

## Discussion

The aim of this investigation was to leverage structural biology, infrared spectroscopy, pharmacological analysis, and MD simulations to pinpoint crucial amino acid residues governing the signaling activity of KOR and to unravel the molecular basis of biased agonism in GPCRs. Alongside the static structural data of G protein-bound KOR acquired with cryo-EM, insights into the conformational changes of KORs preceding G protein binding were gleaned from dynamic infrared spectroscopy, enabling the identification of amino acids involved in signal transduction. K227$^{5.40}$ and Y312$^{7.34}$ were identified as pivotal residues for β-arrestin-recruitment based on cryo-EM data and pharmacological analyses, whereas C286$^{6.47}$ and H291$^{6.52}$ were highlighted through FT-IR and pharmacological findings (Supplementary Fig. 22). Since all the alanine mutants of these identified amino acids showed reduced β-arrestin-recruitment activity, we used MD simulations to elucidate the associated structural changes. This investigation yields promising insights into these amino acids from a drug discovery standpoint.

Previous studies have implicated two amino acids, K227$^{5.40}$ and Y312$^{7.34}$, in the signaling properties of KOR[17,18]. This study adds further clarity to their roles in signaling selectivity. In the U-50,488H-bound KOR structure, the side chain of K227$^{5.40}$ exhibited weak ionic interactions with the side chain of E297$^{6.58}$, whereas, in the nalfurafine-bound KOR structure, it interacted directly with nalfurafine. The K227$^{5.40}$A mutant exhibited a significant decrease in β-arrestin-recruitment activity upon the addition of agonists, and G protein-coupling activity was slightly worse in the U-50,488H-bound form. When we investigated the KOR agonist-induced responses in the E297$^{6.58}$A mutant, we found that substituting E297$^{6.58}$ with alanine did not significantly reduce either G protein dissociation or β-arrestin recruitment upon stimulation with nalfurafine or U-50,488H (Supplementary Fig. 11). This suggests that the effect of K227$^{5.40}$ on β-arrestin recruitment is independent of its interaction with E297$^{6.58}$. This finding contradicts the claim by Daibani et al. that the ionic bond between

K227$^{5.40}$ and E297$^{6.58}$ plays a critical role in β-arrestin recruitment, and that blocking this bond through K227$^{5.40}$'s interaction with the ligand is necessary for biased agonism. Daibani et al. evaluated the signal transduction activity of the E297$^{6.58}$A mutant only in response to nalfurafine and did not report results for U-50,488H. In the present study, no difference in β-arrestin recruitment activity was observed between the wild-type and E297$^{6.58}$A mutant in response to either nalfurafine or U-50,488H. These findings suggest that the ionic bond between K227$^{5.40}$ and E297$^{6.58}$ is not essential for KOR-mediated β-arrestin recruitment in the presence of U-50,488H.

Additionally, analysis of MD simulations in the WT KOR revealed that the distance between the side chain of K227$^{5.40}$ and nalfurafine predominantly peaked at approximately 4 Å or greater, indicating minimal interaction between them (Supplementary Fig. 23). In other words, we found that both U-50,488H and nalfurafine do not strongly interact with K227$^{5.40}$, yet K227$^{5.40}$ remains crucial for β-arrestin recruitment. These results suggest that the side chain of K227$^{5.40}$ is essential for β-arrestin recruitment because it occupies the space between the ligand and TM5. Specifically, the side chain of K227$^{5.40}$ forms a relatively weak van der Waals interaction with U-50,488H and nalfurafine, and this interaction is thought to be key for β-arrestin recruitment by partially restricting the movement of TM5. Structural information and mutational analysis of Y312$^{7.34}$ have provided key insights into the development of biased ligands. Specifically, the Y312$^{7.34}$F mutant displayed reduced β-arrestin-recruitment activity while preserving strong G protein-coupling activity, highlighting the critical role of the hydroxyl group of tyrosine. The distances from the hydroxyl group on the side chain of Y312$^{7.34}$ to U-50,488H and nalfurafine were measured at 3.9 Å and 3.3 Å, respectively (Fig. 3b). KOR-G$_i$ signaling structures indicate that the interaction between these two agonists and the side chain of Y312$^{7.34}$ involves van der Waals forces, and similar interactions are expected between the Y312$^{7.34}$F phenylalanine side chain and the agonists.

Compared to nalfurafine, U-50,488H shows a greater distance from the side chain of Y312$^{7.34}$F, resulting in weaker van der Waals interactions. This is likely why nalfurafine did not impact the G protein signaling activity of the Y312$^{7.34}$F mutant, while U-50,488H did. In other words, for nalfurafine, the distance from the side chain of Y312$^{7.34}$F was sufficient to support G protein binding but less favorable for β-arrestin recruitment. In contrast, the Y312$^{7.34}$A mutant exhibited a significantly increased distance between the side chain and both nalfurafine and U-50,488H, which likely impacted both G protein signaling and β-arrestin-recruitment activities for these ligands. MD simulation results revealed that the Y312$^{7.34}$A side chain did not interact with nalfurafine, as evidenced by distance histogram peaks of approximately 6 Å and 9 Å (Supplementary Fig. 24).

Designing compounds that maintain a greater distance between the side chain of Y312$^{7.34}$ and nalfurafine is expected to preserve G protein-coupled activity while minimizing β-arrestin recruitment. Che et al. reported the importance of Y312$^{7.34}$ in biased activity, as they observed biased activity with biased ligands of MOR by replacing Y312$^{7.34}$ of KOR with a tryptophan residue. In this study, we elucidated the structures of two KOR-selective agonist binding states and conducted a detailed structural and pharmacological comparison. We found that the interaction of the ligand with Y312$^{7.34}$ is pivotal for biased activity. Zhuang et al. determined the structures of several biased/balanced agonist-bound MOR-G protein complexes and suggested the importance of reducing interaction with TM6/7 in the design of G-protein-biased agonists[28]. This proposal for MOR contradicts the findings of our study, where it is crucial for G-protein-biased agonists to interact appropriately with Y312$^{7.34}$ at TM7 of KOR. Therefore, to develop analgesics without the side effects of KOR, it is imperative to design agents that differ from those targeting MOR.

In contrast, the involvement of C286$^{6.47}$ and H291$^{6.52}$ of TM6 in signaling selectivity, as discovered through FT-IR measurements and

mutant analysis in this study, has not been previously documented. Research on the signal selectivity of MOR and DOR does not address these specific amino acids. The available information suggests that C286[6.47] is part of the CWxP motif, while H291[6.52] interacts with the side chain of W287[6.48] during ligand binding and functions as a component of the toggle switch[29]. Structural analysis and FT-IR results from our study suggest that these amino acids do not directly interact with the agonist but rather modify the interaction network with other amino acids in TM7 upon agonist binding. Furthermore, the alanine mutants exhibit diminished β-arrestin recruitment, indicating the significance of the interaction between these amino acids in TM6 and TM7 for biased signaling (Supplementary Fig. 22). Specifically, the interaction between C286[6.47] and T321[7.43] is crucial for β-arrestin-recruitment activity, while it does not influence G protein-coupling. Hence, the design of agonists or allosteric ligands that disrupt these interactions could potentially lead to the development of KOR agonists devoid of side effects.

H291[6.52] forms an intricate interaction network with N322[7.44] via W287[6.48], and the reduced β-arrestin-recruitment activity observed in the H291[6.52]A mutant may result from the disruption of these networks. In essence, the H291[6.52]A mutation could alter the orientation of the side chain of W287[6.48], consequently affecting the interaction between W287[6.48] and N322[7.44] of TM7.

MD simulations of alanine mutants of the amino acids identified as important for β-arrestin recruitment revealed that each amino acid plays a critical role in maintaining the conformational states required for β-arrestin-recruitment activity (Fig. 5). In summary, this study successfully identifies the amino acid residues crucial for β-arrestin recruitment, as well as those that interact with agonists or whose interaction networks are modified upon agonist binding (Supplementary Fig. 22).

This study demonstrated that nalfurafine exhibits balanced agonist properties. However, some previous reports have described G protein-biased signaling. Specifically, studies have reported that the signaling profile of nalfurafine is G protein-biased[17,30–32], unbiased[33], or β-arrestin-biased[34], indicating a lack of clear consensus regarding its bias characteristics. In some studies, reporting G-protein bias, the magnitude of the observed bias is relatively modest[31,32]. To date, there are few known examples of potent G protein-biased agonists for the KOR, and their development remains highly anticipated. Our integrated approach, amalgamating structural analysis and infrared spectroscopy, elucidates a pivotal signaling mechanism crucial for drug discovery. This methodology, directly applicable to opioid receptors like KORs, holds promise for extension to other GPCRs. Approaches akin to this study show potential in facilitating the rational design and enhancement of therapeutics, encompassing biased agonists for KORs, and can be extrapolated to the development of biased agonists for numerous other GPCRs.

## Methods
### Cells
*Spodoptera frugiperda* 9 (Sf9) insect cells were cultured in ESF-921 (Wako), 50 units/mL penicillin, 50 μg/mL streptomycin (Wako), and 0.5 μg/mL amphotericin B at 27 °C. Parental human embryonic kidney 293 (HEK293) cells[35] were grown in Dulbecco's modified Eagle's medium (DMEM; Nacalai Tesque) supplemented with 10% FBS (Nichirei Biosciences), 100 units/mL penicillin, and 100 μg/mL streptomycin (Nacalai Tesque) at 37 °C in a 5% CO₂ incubator.

### Constructs
For the human KOR construct, the N-terminal (1–53) and C-terminal (359–380) regions were deleted from the wild-type KOR based on the previously reported crystal structure of nanobody-stabilized KOR[17,18], and the deleted N-terminal (1–53) was replaced with b562RIL (BRIL). An N-terminal A 3C protease site was inserted between BRIL and KOR, and

a 3C protease site was also inserted at the C-terminus of KOR to introduce GFP and 10 × His tag. The sf9 expression vector for the heterotrimeric G protein was provided by Brian K Kobilka's lab, with $G_i$ introduced into pFastBac and $G_β$ and $G_γ$ into the pFastBac Dual vector. Sf9 expression vector with scFv16 was introduced into the pFastBac vector and provided by Nureki Lab[36]. The primers used for construct generation and mutant preparation were custom-synthesized by Thermo Fisher Scientific, and their sequences are provided in the Source Data file.

### Expression and purification of the human KOR-$G_{i1}G β_1 γ_2$-scFv16 complex
Recombinant baculoviruses were produced using the Bac-to-Bac system (Invitrogen) and infected Spodoptera frugiperda (Sf9) cells. Briefly, Sf9 cells were cultured in ESF921 medium at a density of $3–4 × 10^6$ cells/ml and co-infected with three different baculoviruses: KOR, $G_{i1}$, and $G_{β1γ2}$ in a 1:2:1 ratio. 27 °C, 125 rpm, 48–72 h. After infection for 2 h, cells were collected by centrifugation, and cell pellets were stored at −80 °C. Insect cell membranes were resuspended in suspension buffer containing 30 mM HEPES-NaOH (pH 7.5), 750 mM NaCl, 5 mM imidazole, 1 mg/ml iodoacetamide, 1 μM agonist, 10 mM leupeptin, 1 mM benzamidine, and apyrase. The suspension was incubated at 4 °C for 30 min; DDM and CHS were added at final concentrations of 1% (w/v) and 0.2% (w/v), respectively, and incubation continued for another 2 h. The supernatant was ultracentrifuged at $100,000 × g$ for 30 min and purified with Ni-NTA Superflow resin (Qiagen). The purified supernatant was applied to a column (GFPNb column) conjugated with a GFP recognition antibody (GFP nanobody) and washed with 3 volumes of buffer (30 mM HEPES-NaOH (pH 7.5), 750 mM NaCl, 0.1% (w/v) DDM, 0.03% (w/v) CHS, 10 μM (agonist) and washed. DDM was then gradually replaced over 1 h with buffer containing 0.01% (w/v) lauryl maltose neopentyl glycol (MNG), and the NaCl concentration was lowered to 100 mM. KOR-$G_iG_{bg}$ complex was eluted by adding 3C protease in the presence of 10 μM agonist. ScFv16 was purified as in the previously reported study of the structural analysis of the MT1- $G_iG_{bg}$ -scFv16 complex[36]. The preparation of the GPCR-G protein complex using the GFPNb column was similar to our previously reported method for the prostaglandin receptor-$G_iG_{bg}$ complex[37]. The purified KOR-$G_iG_{bg}$ complex and scFv16 were combined in a 1:2 molar ratio and incubated overnight at 4 °C. The resulting KOR-$G_iG$-scFv16 complex was purified using a Superdex 200 Increase 10/300 GL column (Cytiva) in 20 mM HEPES-NaOH (pH 7.5), 100 mM NaCl, 0.001% (w/v) MNG, 0.0003% GDN, 0.001% (w/v) CHS, 1 μM agonist, and further purified using size exclusion chromatography in a buffer of 100 mM TCEP (Supplementary Fig. 1a–d). The eluted peak fraction was concentrated to approximately 5 mg/ml using an Amicon Ultra concentrator (Millipore) with a molecular weight cutoff of 100 kDa.

### Cryo-EM grid preparation and data collection
A 3 μl aliquot of sample solution was applied to a glow discharge QUANTIFOIL R1.2/1.3 Au 300 mesh grid (Quantifoil Micro Tools GmbH) on a Vitrobot Mark IV (Thermo Fisher Scientific). After wiping off the excess solution on the grid with filter paper, the samples were rapidly frozen in liquid ethane. The frozen grids were screened using a Talos Glacios cryo-transmission electron microscope (cryo-TEM) (operated at 200 keV and equipped with a Falcon 4 direct electron detector (Thermo Fisher Scientific)) at Institute for Life and Medical Sciences, Kyoto University, to check sample conditions such as ice thickness and particle dispersion. Cryo-EM data collection was performed using a Titan Krios cryo-TEM (Thermo Fisher Scientific) in EFTEM nanoprobe mode operating at 300 keV equipped with a Cs corrector (CEOS GmbH) at the Institute for Protein Research, Osaka University. Images were acquired as movies using a Gatan BioQuantum energy filter with a slit width of 20 eV and a K3 direct detection camera

(Gatan, Inc). For U-50,488H or nalfurafine-bound KOR-$G_i$ complexes, a total of 18,885 and 14,349 movies were collected, respectively, at a pixel size of 0.675 $Å^2$ and a total dose of 60 $e^-/Å^2$ Automated data collection was performed using a 3 × 3 hole-pattern beam-image-shift scheme with a nominal defocus range of −0.7 to −1.5 μm facilitated by the SerialEM software[38].

## Image processing
Image processing was performed with cryoSPARC[39]. Micrographs were motion-corrected using patch motion correction, and CTF parameters were estimated using patch CTF estimation. For U-50,488H- or nalfurafine-bound KOR datasets, particles were initially picked using a blob picker (minimum particle size 100, maximum particle size 200). Subsequently, particle extraction was performed with a particle box size of 360 pixels, resulting in the extraction of 4,148,764 particles from the U-50,488H-bound KOR dataset and 7,066,947 particles from the nalfurafine-bound KOR dataset. Only in the case of U-50,488H-bound KOR, once 2D classification was done, Topaz extraction was performed, and 4,148,764 particles were extracted from 18,845 micrographs. After multiple rounds of 2D classification, we selected 3,078,397 or 3,957,130 particles for ab-initio reconstruction. In the case of the data of the nalfurafine-bound KOR, a set of 2,439,864 particles was further classified into two classes using ab-initio reconstruction followed by heterogenous refinement. Particles from one class (1,225,096 or 858,423 particles) were combined and subjected to homogeneous refinement and non-uniform refinement, yielding a resolution of 2.90 Å and 2.76 Å for U-50,488H- and nalfurafine-bound KOR-$G_i$ complex, respectively. Resolution is calculated using the "gold standard" Fourier Shell Correlation (FSC) between two independently refined maps reconstructed independently from randomly divided half datasets. The image processing steps are summarized in Supplementary Figs. 2 and 3.

## Model building and refinement
The atomic model building was performed by manual iterative building in Coot[40], followed by refinement with phenix.real_space_refine in the Phenix program suite[41]. In this model, 97% of the residues were in the favored regions of the Ramachandran plot, and all the others were in the allowed regions. Refinement statistics are shown in Supplementary Table 3.

## Preparation of proteoliposome of KOR
For ATR-FTIR spectroscopy measurements, asolectin lipid pre-sonicated to form unilamellar structures was added to the purified KOR (final DDM concentration: 0.01%) at a molar ratio of 1:10 (KOR:lipid), followed by rotational mixing at 4 °C for 2 h. Subsequently, BioBeads were added to remove detergent, and the mixture was incubated for another 2 h to complete the reconstitution process. After removing the Biobeads, the reconstituted KOR was recovered by ultracentrifugation. Following several cycles of washing/spinning, the lipid-reconstituted KOR was suspended in a buffer consisting of 5 mM phosphate (pH 7.5) and 10 mM KCl. Furthermore, the determination of detergent removal was based on the infrared absorption peaks attributed to detergent (observed in the 1200 $cm^{-1}$ region) in the ATR-FTIR measurements.

## Ligand binding-induced ATR-FTIR difference spectroscopy
A 3 μL suspension of lipid-reconstituted KOR was dropped onto the surface of a silicon ATR crystal (with 3 internal reflections). After gently drying to form a thin film sample by natural drying, the film sample was rehydrated through a flow cell maintained at 20 °C by circulating water at a flow rate of 0.4 mL $min^{-1}$, with a solvent containing 200 mM phosphate buffer (pH 7.5) containing 140 mM KCl and 3 mM $MgCl_2$.

Before measurements, 0.1 mM naltrexone was perfused into the buffer to bind KOR with naltrexone. Using an FTIR spectrometer equipped with a liquid nitrogen-cooled MCT detector (Bio-Rad FTS6000, Agilent, CA, USA), ATR-FTIR absolute absorbance spectrum of the inactive form of KOR bound to Naltrexone was first recorded at a resolution of 2 $cm^{-1}$ (averaging 64 spectra). Next, ATR-FTIR absolute absorbance spectrum was recorded in a second buffer solution containing 0.01 mM nalfurafine or U-50,488H added, during which ligands binding to KOR were exchanged.

The ATR-FTIR difference absorbance spectrum (hereafter referred to as difference spectrum) was obtained by subtracting the absolute absorbance spectrum of the naltrexone-bound form from the absolute absorbance spectrum of the nalfurafine or U-50,488H-bound form. Spectral corrections were made for protein swelling/shrinkage, water vapor, $CO_2$, and contributions from water/buffer components in the spectra.

## Plasmids
For the NanoBiT-G protein dissociation assay, the full-length human KOR was inserted into the pCAGSS expression vector with the N-terminal fusion of the hemagglutinin-derived signal sequence (ssHA), FLAG epitope tag and a flexible linker (MKTIIALSYIFCLVFA-DYKDDDDKGGSGGGGSGGSSSGGG; the FLAG epitope tag is underlined). The resulting construct was named ssHA-FLAG-KOR. For the NanoBiT-β-arrestin recruitment assay, the ssHA-FLAG-KOR construct was C-terminally fused with the flexible linker, and the SmBiT fragment GGSGGGGSGGSSSGGVTGYRLFEEIL; the SmBiT is underlined). The resulting construct was named ssHA-FLAG-KOR-SmBiT.

## NanoBiT-G protein-dissociation assay
KOR ligand-induced G protein dissociation was measured by the NanoBiT-G protein dissociation assay[42], in which the interaction between a Gα subunit and a Gβγ subunit was monitored by the NanoBiT system (Promega). Specifically, a NanoBiT-$G_i$ protein consisting of the $Gα_{i1}$ subunit fused with a large fragment (LgBiT) at the α-helical domain (between the residues 91 and 92 of $Gα_{i1}$; $Gα_{i1}$-LgBiT) and the N-terminally small fragment (SmBiT)-fused $Gγ_2$ subunit with a C68S mutation (SmBiT-$Gγ_2$-CS) was expressed along with untagged $Gβ_1$ subunit. HEK293A cells (Thermo Fisher Scientific, cat no. R70507) were seeded in a 6-well culture plate at a concentration of $2 × 10^5$ cells $ml^{-1}$ (2 ml per well in DMEM (Nissui) supplemented with 5% fetal bovine serum (Gibco), glutamine, penicillin, and streptomycin), one day before transfection. Transfection solution was prepared by combining 6 μL (per dish hereafter) of polyethylenimine (PEI) Max solution (1 mg $ml^{-1}$; Polysciences), 200 μL of Opti-MEM (Thermo Fisher Scientific), and a plasmid mixture consisting of 200 ng ssHA-FLAG-KOR (or an empty plasmid for mock transfection), 100 ng $Gα_{i1}$-LgBiT, 500 ng $Gβ_1$ subunit, and 500 ng SmBiT-$Gγ_2$-CS subunit. To compare expression-matched KOR mutants, the WT KOR plasmid was serially titrated. After incubation for 1 day, the transfected cells were harvested with 0.5 mM EDTA-containing Dulbecco's PBS, centrifuged, and suspended in 2 ml of HBSS containing 0.01% bovine serum albumin (BSA; fatty acid-free grade; SERVA) and 5 mM HEPES (pH 7.4) (assay buffer). The cell suspension was dispensed in a white 96-well plate at a volume of 80 μL per well and loaded with 20 μL of 50 μM coelenterazine (Angene) diluted in the assay buffer. After a 2 h incubation at room temperature, the plate was measured for baseline luminescence (SpectraMax L, Molecular Devices), and titrated concentrations of a KOR ligand (nalfurafine or U-50,488H; 20 μL; 6× of final concentrations) were manually added. The plate was immediately read for the second measurement as a kinetics mode and luminescence counts recorded from 5 to 10 min after compound addition were averaged and normalized to the initial counts. The fold-change values were further normalized to those of vesicle-treated samples and used to plot the G protein dissociation response. Using the Prism 9 software (GraphPad Prism), the G-protein dissociation signals were fitted to a four-parameter sigmoidal concentration-response curve with a

constraint of the *HillSlope* to absolute values less than 2. For each replicate experiment, the parameters *Span* (= *Top – Bottom*), pEC$_{50}$ (negative logarithmic values of EC$_{50}$ values), and *Span*/EC$_{50}$ of the individual KOR mutants were normalized to those of WT KOR performed in parallel, and the resulting parameters $E_{max}$, $\Delta$pEC$_{50}$ and relative intrinsic activity (RAi), respectively, were used to denote ligand response activity of the mutants.

## NanoBiT-β-arrestin recruitment assay
Ligand-induced β-arrestin recruitment assay was measured as described previously[43]. Transfection was performed according to the same procedure as described in the "NanoBiT-G protein-dissociation assay" section, except for a plasmid mixture consisting of 500 ng ssHA-FLAG-KOR-SmBiT and 100 ng N-terminally LgBiT-fused β-arrestin2. The transfected cells were dispensed into a 96-well plate, and ligand-induced luminescent changes were measured by following the same procedures as described for the NanoBiT-G protein-dissociation assay.

## Flow cytometry
Transfection was performed according to the same procedure as described in the "NanoBiT-G protein-dissociation assay" and the "NanoBiT-β-arrestin recruitment assay" sections. One day after transfection, the cells were collected by adding 200 μl of 0.53 mM EDTA-containing Dulbecco's PBS (D-PBS), followed by 200 μl of 5 mM HEPES (pH 7.4)-containing Hank's Balanced Salt Solution (HBSS). The cell suspension was transferred to a 96-well V-bottom plate in duplicate and fluorescently labeled with an anti-FLAG epitope (DYKDDDDK) tag monoclonal antibody (Clone 1E6, FujiFilm Wako Pure Chemicals; 10 μg per ml diluted in 2% goat serum- and 2 mM EDTA-containing D-PBS (blocking buffer)) and a goat anti-mouse IgG secondary antibody conjugated with Alexa Fluor 488 (Thermo Fisher Scientific, 10 μg per ml diluted in the blocking buffer). After washing with D-PBS, the cells were resuspended in 200 μl of 2 mM EDTA-containing-D-PBS and filtered through a 40 μm filter. The fluorescent intensity of single cells was quantified by an EC800 flow cytometer equipped with a 488 nm laser (Sony). The fluorescent signal derived from Alexa Fluor 488 was recorded in an FL1 channel, and the flow cytometry data were analyzed with the FlowJo software (FlowJo). Live cells were gated with a forward scatter (FS-Peak-Lin) cutoff at the 390 setting, with a gain value of 1.7. Values of mean fluorescence intensity (MFI) from approximately 20,000 cells per sample were used for analysis. For each experiment, we normalized an MFI value of the mutants by that of WT performed in parallel and denoted relative levels.

## Saturation binding assay
Saturation binding assays with [³H] U-69,593 were performed in binding buffer (50 mM Tris, 1 mM EDTA, 5 mM MgCl₂, pH 7.4 adjusted with HCl) to determine the equilibrium dissociation constant ($K_d$). [³H] U-69,593 (Revvity, Inc., Waltham, MA, USA) was diluted with binding buffer to prepare working solution ranging from 0.5 to 12 nM. Expi293F cells (Thermo Fisher Scientific, catalogue number A14527) were cultured to a density of $3 \times 10^6$ cells/mL in 50 mL of Expi293 medium supplemented with penicillin and streptomycin. The transfection solution was prepared by mixing 200 μL of polyethyleneimine (PEI) Max solution (1 mg/mL; Polysciences), 5.0 mL of Opti-MEM (Thermo Fisher Scientific), and 40 μg of the wild-type or mutant (Y312A, Y312F) KOR plasmid. This mixture was then added to the Expi293F cells, which were incubated at 37 °C for 2 days. The cells were collected by centrifugation, lysed by osmotic shock, and the membrane fraction was isolated by ultracentrifugation. The resulting pellet was then resuspended in PBS. The membrane fraction was prepared using the same procedure as the saturation binding assay. The membrane suspension was then incubated with various concentrations of [³H]-U69,593 in 250 μL of binding buffer at 25 °C for 2 h with gentle agitation at 300 rpm. To define nonspecific binding, 10 μM unlabeled U-69,593 was co-incubated. The reaction was terminated by filtration using a Filtermat B glass filter (PerkinElmer Co., Ltd.) with a FilterMate cell harvester (PerkinElmer Co., Ltd.). Filters were washed three times with 50 mM Tris buffer and dried at 60 °C for 70 min. Subsequently, MeltiLex B/HS (PerkinElmer Co., Ltd.) was melted onto the dried filters at 90 °C for 5 min. The radioactivity retained in the filters was measured with a Microbeta scintillation counter (PerkinElmer Co., Ltd.). Specific binding was calculated as the difference between total binding and nonspecific binding. $K_d$ values were determined using a "one site–specific binding" model and analyzed with Prism software (version 8.4.3; GraphPad Software Inc., La Jolla, CA, USA). The data points were represented as the mean values ± S.E.M., and a saturation curve was fitted to the data, with 95% CI also calculated from at least three independent experiments, each performed in triplicate.

## Competitive binding assay
To determine inhibitory constant ($K_i$) of the test compounds, competitive binding assays were performed. The membrane preparation of KOR-expressing cells was carried out in the same manner as described for the saturation binding assay mentioned above. Membrane suspension obtained from HEK293 cells stably expressing either WT KOR or mutant KOR mutants (Y312A, Y312F) was incubated in 250 μL binding buffer containing various concentrations of KOR agonist and [³H]-U69,593 at 25 °C for 2 h with gentle shaking at 300 rpm. The concentration of [³H] U-69,593 was adjusted according to the $K_d$ values for each membrane type. To define nonspecific binding, 10 μM unlabeled U-69,593 was co-incubated. The reaction was terminated by filtration using a Filtermat B glass filter (PerkinElmer Co., Ltd.) with a FilterMate cell harvester (PerkinElmer Co., Ltd.). Filters were washed three times with 50 mM Tris buffer and dried at 60 °C for 70 min. Subsequently, MeltiLex B/HS (PerkinElmer Co., Ltd.) was melted onto the dried filters at 90 °C for 5 min. The radioactivity retained in the filters was measured with a Microbeta scintillation counter (PerkinElmer Co., Ltd.). Specific binding was calculated as the difference between total binding and nonspecific binding. Sigmoidal concentration-response curves and $K_i$ values were calculated according to the Cheng–Prusoff equation using the GraphPad Prism software.

## MD simulation for KOR wild-type and K227$^{5.40}$, C286$^{6.47}$, H291$^{6.52}$, and Y312$^{7.34}$ mutants
We performed three independent one-microsecond all-atom MD simulations for each of the wild-type (WT) κ-opioid receptor (KOR) and four mutants—K227$^{5.40}$A, C286$^{6.47}$A, H291$^{6.52}$A, and Y312$^{7.34}$A—totaling 15 microseconds of simulation time. The initial structures were based on the cryo-EM structure of the KOR–nalfurafine complex. The G protein was excluded from the cryo-EM structure to investigate the intrinsic dynamics of the intracellular region. Protonation states were assigned using the Adaptive Poisson-Boltzmann Solver (APBS)-PDB2PQR module at pH 7.0[44]. The conserved D105$^{2.50}$ was protonated as this state is commonly adopted in active GPCRs. Missing regions in the KOR–nalfurafine complex were modeled using MODELLER[45,46].

Residue substitutions and system setup with lipid bilayers were performed using CHARMM-GUI. The KOR was embedded in a pure 1-palmitoyl-2-oleoyl-sng-lycero-3-phosphocholine (POPC) membrane consisting of approximately 160 lipids per leaflet, resulting in a membrane area of ~12 × 12 nm and a system height of 12.8 nm. This membrane model serves as a simplified representation of human cell membranes, with POPC chosen due to its common presence in Biomembranes. Each system was solvated with ~31,000 TIP3P water molecules, and NaCl was added to neutralize the system charge and achieve a salt concentration of 150 mM. The final systems included the KOR, nalfurafine, lipids, water molecules, sodium ions, and chloride ions.

MD simulations were conducted using the AMBER software, employing the AMBER ff19SB force field for proteins, Lipid21 force field for lipids, and the general AMBER force field (GAFF) for ligands. The ligand was modeled with a neutral net charge.

Periodic boundary conditions were applied in a rectangular box. A time step of 2 fs was used, and trajectories were recorded every 10 ps. Temperature was maintained at 310 K using a Langevin thermostat, and pressure was controlled at 1 bar using a Monte Carlo barostat. The SHAKE algorithm was employed to constrain bonds involving hydrogen atoms. Short-range electrostatic and van der Waals interactions were calculated with a cutoff distance of 1.2 nm.

The simulation protocol began with energy minimization under constant volume without pressure control. The system was then heated to 100 K over 20 ps while applying harmonic restraints of 10.0 kcal/(mol Å$^2$) to the KOR–nalfurafine complex and 2.5 kcal/(mol Å$^2$) to the lipid bilayer. Subsequently, the temperature was gradually increased to 310 K over 400 ps with the same restraints. A Langevin thermostat and an anisotropic Berendsen weak-coupling barostat were used during heating and pressure equilibration. A 4-ns equilibration followed, during which the restraints on the KOR–nalfurafine complex and lipid bilayer were gradually reduced to allow the system dimensions and density to stabilize. Finally, production MD simulations were conducted under constant temperature and pressure using the Langevin thermostat and a semi-isotropic Monte Carlo barostat. In total, we performed 15 independent one-microsecond MD simulations. Interaction fingerprint analysis was conducted using ProLIF with default values (cutoff 6 Å)[24].

### Quantification and statistical analysis

In the signaling assay, maximum luminescence intensity post stimulation was quantified. The luminescence intensity reached a plateau about 10 min after stimulation. Each point represents the mean value ± s.e.m. All the measurements were performed in triplicate. Sigmoid curve fitting was performed with Prism 7 (GraphPad).

### Lead contact

Further information and requests for resources and reagents should be directed to and will be fulfilled by the lead contact, Ryoji Suno (suno.ryo@kmu.ac.jp) and Takuya Kobayashi (kobayatk@hirakata.kmu.ac.jp).

### Reporting summary

Further information on research design is available in the Nature Portfolio Reporting Summary linked to this article.

## Data availability

The cryo-EM data generated in this study have been deposited in the Electron Microscopy Data Bank (EMDB) under accession codes EMDB-65622 [https://www.ebi.ac.uk/emdb/EMD-65622] and EMDB-64947 [https://www.ebi.ac.uk/emdb/EMD-64947]. The coordinate data generated in this study have been deposited in the Protein Data Bank (PDB) under accession codes PDB 9V6O and PDB 9W49. The molecular dynamics (MD) simulation data generated in this study are provided in the Source Data file. Source data are provided with this paper.

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

## Acknowledgements

The plasmids for expressing scFv16 were provided by Hiroyuki Okamoto (Tokyo University). The authors thank Kayo Sato, Shigeko Nakano, and Ayumi Inoue at Tohoku University for their assistance in plasmid preparation. This research was supported by the Takeda Science Foundation (to R.Suno, Y.S., and T. Kobayashi), AMED Core Research for Evolutional Science and Technology (CREST) (JP21gm0910007 to T. Kobayashi), AMED Science and Technology Platform Program for Advanced Biological Medicine (JP21am0401020 to T. Kobayashi), AMED Research on Development of New Drugs (JP20ak0101103 to T. Kobayashi), AMED Moonshot Research and Development Program (JP21zf0127005 to T.S.), the Naito Foundation (to T. Kobayashi), Koyanagi Foundation (to T. Kobayashi), the Japan Foundation for Applied Enzymology (16H007 to T.S.); JST FOREST Program JPMJFR215T (to A.I.), JST Moonshot Research and Development Program (JPMJMS2023 to A.I.), JST SPRING (to A.T.), JST Mirai Program JPMJMI22H5 (to T.S., A.I.), JSPS KAKENHI (19H03428 and 24K02231 to R. Suno.; 21H04791, 21H05113, and JPJSBP120218801 to A.I.; 20H03230 to A.M.; 23KJ1997 to S.Y.), Grant-in-Aid for Transformative Research Areas (21H05111 to R. Suno., A.I., and T.S.; 21H05112 to R. Suno; 21H05113 to A.I.; 21H05115 to T.S.), MEXT LEADER Program (to Y.S.), the Young Runners in Strategy of Transborder Advanced Researches (TRiSTAR) program (to T.S.), and Grant-in-Aid for Specially Promoted Research (JP21H04969 to H.K). Cryo-EM analysis was supported by the Cooperative Research Program (Joint Usage/Research Center program) of Institute for Life and Medical Sciences, Kyoto University, and the Platform Project for Supporting Drug Discovery and Life Science Research (Basis for Supporting Innovative Drug Discovery and Life Science Research (BINDS)) from AMED (JP25121001 to T. Kato).

## Author contributions

C.S. carried out protein expression and purification of the receptor, G proteins, GFP nanobody, and scFv16 fragment. T.S. and H.N. synthesized the compounds. C.S., T.T., Akitoshi Inoue, E.A created a mutant construct of KOR. C.S. prepared the cryo-EM sample of the KOR-Gi-scFv16 complex. C.S., R. Suno, Y.S., and M.H. carried out the cryo-EM data collection. R. Suno carried out cryo-EM data processing and model building of the KOR-Gi-scFv16 complex. R. Suzuki, T.O., and R. Kise carried out the signaling assays. R.N. and S.I. performed the infrared spectroscopy measurements. S.Y. and A.M. performed and analyzed MD simulations. A.T., R. Katamoto, and T.S. conducted the ligand binding experiments. R. Suno designed the project, and Asuka Inoue, H.K., K.K., T. Kato, and T. Kobayashi supervised the overall project. C.S. and R. Suno wrote the manuscript. All authors discussed the results and commented on the manuscript.

## Competing interests

## Additional information

[1]Department of Medical Chemistry, Kansai Medical University, Hirakata, Japan. [2]Department of Life Science and Applied Chemistry, Nagoya Institute of Technology, Showa-ku, Nagoya, Japan. [3]Graduate School of Pharmaceutical Sciences, Kyoto University, Kyoto, Japan. [4]Department of Physics, School of Science and Technology, Meiji University, Kawasaki, Kanagawa, Japan. [5]International Institute for Integrative Sleep Medicine (WPI-IIIS), Tsukuba Institute for Advanced Research (TIAR), University of Tsukuba, Tsukuba, Ibaraki, Japan. [6]Graduate School of Pharmaceutical Sciences, Tohoku University, Sendai, Miyagi, Japan. [7]Institute for Protein Research, Osaka University, Suita, Japan. [8]Hakubi Center for Advanced Research, Kyoto University, Kyoto, Japan. [9]Institute for Life and Medical Sciences, Kyoto University, Kyoto, Japan. [10]Graduate School of Biostudies, Kyoto University, Kyoto, Japan. [11]Faculty of Medicine, University of Tsukuba, Tsukuba, Ibaraki, Japan. [12]OptoBioTechnology Research Center, Nagoya Institute of Technology, Showa-ku, Nagoya, Japan. [13]Japan Agency for Medical Research and Development (AMED), Core Research for Evolutional Science and Technology (CREST), Chiyoda-ku, Tokyo, Japan. [14]These authors contributed equally: Chiyo Suno-Ikeda, Ryo Nishikawa, Riko Suzuki. ✉e-mail: kobayatk@hirakata.kmu.ac.jp; suno.ryo@kmu.ac.jp

