## [Transparent Peer Review file · Nature Communications]

Structural and Dynamic Insights into the Biased Signaling Mechanism of the Human Kappa Opioid Receptor

Corresponding Author: Dr Ryoji Suno

Version 0:

Reviewer comments:

Reviewer #1

(Remarks to the Author)

This work seeks to elucidate the structural dynamics of human kappa-opioid receptor (KOR)-Gi signaling complex as well as the inactive form (without Gi) upon interaction with a G-protein-biased agonist and a balanced agonist. The authors have used cryo-EM and ATR-FTIR spectroscopy to shed light on the structural details of ligand binding by the lipid membrane reconstituted receptor. Further cell-based studies on the receptor incorporating point mutations allowed identification of the key amino acids involved in ligand binding and arresting recruitment. The work appears to be scientifically and technically sound and useful for developing better KOR-mediated therapies. This review focuses on the ATR-FTIR part only.

ATR-FTIR procedures:

“KOR solubilized in detergent was reconstituted into asolectin liposomes with a 10-fold molar excess.” What detergent was used? Were the liposomes unilamellar made by extrusion or sonication? Was the detergent/lipid molar ratio 10:1?

Why was 200 mM phosphate buffer used? Together with 140 mM KCl it will produce a very high ionic strength.

The 0.1 mM concentration of naltrexone is too high, certainly not physiologically relevant. Was the sample flushed with buffer to remove unbound naltrexone before FTIR spectral measurement?

What is “absolute absorbance” and how is it different from just absorbance? What reference (background) was used for the absorbance spectra? Please explain “Corrections were made for protein/lipid contraction.”

Is there evidence that injection of 0.01 mM Nalfurafine or U-50,488 replaces the receptor-bound naltrexone? If the affinity of naltrexone is higher, in addition to its 10-fold higher concentration, the exchange may be far from complete.

ATR-FTIR data and interpretation

Lines 279-280: “...distinct changes in the infrared signals emerged at amide-I of an α -helix (1,700–1,600 cm^{-1}).” This reads like the whole 1,700–1,600 cm^{-1} region is the alpha helical region. Consider “...distinct spectral changes in the 1666 cm^{-1} –1650 cm^{-1} region of the amide I band have been detected and ascribed to conformational changes in the alpha helices of the receptor.”

Lines 270-282: “C-N stretching of histidine imidazole (1,200–1,100 cm^{-1}), and S-H stretching of cysteine side chain (2,600–2,500 cm^{-1}).” Please provide references for these wavenumbers of His and Cys side chains.

Legend to Figure 4: “Signals from nalfurafine and U-50,488H were observed as they interacted with the KOR in the presence of the antagonist naltrexone.” Please explain if both agonists bind to the same site and also if the antagonist binds to same site as the agonists. Do the agonists displace the antagonist? (Is it competitive binding to the same site?)

Lines 283-284: How does a decrease in the absorbance intensity of the Cys side chain at 2550 cm^{-1} indicate a difference in

the hydrogen-bonding environment of cysteine?

Lines 284-287: Can you interpret the possible helix perturbation as judged from the amide I spectral changes, i.e. the shift from 1651 cm⁻¹ to 1666-1663 cm⁻¹? Have you considered an alpha-I to alpha-II helix transition?

Reviewer #2

(Remarks to the Author)

In this study, Suno-Ikeda et al. investigates the structural and dynamic insights into the biased signaling mechanism of the human kappa opioid receptor (KOR) by multiple techniques. The authors determined the cryo-EM structures of KOR-Gi signaling complexes bound with the balanced agonist U-50,488H and the G protein-biased agonist nalfurafine, which provided valuable insights into the ligand-binding modes and conformational changes associated with biased signaling. Structural comparison and mutagenesis studies suggested the important role of the interaction of the ligand with K2275.40 and Y3127.34 in mediating G protein bias. Complementing the static structural data from cryo-EM, the authors investigated the ligand-dependent dynamic changes in the amino acid side chains of KOR prior to G protein binding through a novel use of vibrational spectroscopy, which identified another two specific residues, C2866.47 and H2916.52, played crucial roles in G protein versus arrestin recruitment. Overall, this manuscript presents a significant and well-executed study that advances our understanding of the biased signaling of KOR. My comments:

Major concerns:

1. Calculating the bias factor is a widely used method in pharmacological studies of GPCRs to determine whether a ligand behaves as a biased agonist. Since nalfurafine is not a purely G-protein biased KOR agonist, the authors are encouraged to characterize the bias properties of the compounds used in this study by their established system, using the endogenous ligand dynorphin as the reference.
2. For the nalfurafine-bound KOR-Gi complex, the authors state in line 127 that they obtained four different Gi coupling states of KOR. Do these four coupling models correspond to different KOR conformations when bound to nalfurafine? In a previous study reporting the crystal structure of nalfurafine-bound KOR-Nb39, three active-state KOR conformations were observed during molecular simulations (El Daibani, A., et al. (2023). *Nat Commun*: 14(1), 1338.). Is there any relationship between the KOR conformations observed in this study and those reported previously?
3. The legends for the figures presenting pharmacological data, such as Figures 2, 3, Supplemental Figures 7 and 9, are not well-explained. After examining the entire manuscript, it is still unclear what WT (1:1), (1:2), ..., (1:16) mean and why the authors set these references. Furthermore, only the response curves for Gi and arrestin recruitment for some KOR mutants include two WT references, WT (1:1) and (1:16), which is confusing. Please clarify these points.
4. In line 202, the authors state that the K2275.40-E2976.58 interaction is critical for arrestin activity of KOR. Does the KOR-E2976.58A mutant, which also disrupts this polar contact, exhibit G-protein biased activity when activated by U-50,488H? Additionally, the K2275.40-E2976.58 polar interaction is absent in the dynorphin-bound KOR structure (*Cell*, 186(2), 413-427). Does this mean that the bias determinant K2275.40 is ligand-dependent? The authors should discuss this point further.
5. Since some KOR mutants influence not only the EC₅₀ but also the E_{max} value of Gi and arrestin signaling, it would be better for the authors to calculate and compare the bias factors of KOR mutants for each ligand. This would demonstrate the mutational effect of each residue on the biased signaling of KOR more directly, especially for those residues emphasized by the authors, such as Q1152.60, K2275.40, Y3127.34, C2866.47, and H2916.52.
6. Please show the local densities of K2275.40 and E2976.58 in Figure 2a, Y3127.34 and I2946.55 in Figure 2d, and C2866.47 and T3217.43 in Figure 4d.
7. The authors have provided a thorough structural analysis and mutagenesis data. However, it would be valuable to include molecular dynamics simulations or computational modeling to further support the proposed interaction networks and conformational changes associated with biased signaling, such as K2275.40A, Y3127.34A and C2866.47A mutants.

Minor:

1. Line 44: the words "responsible for" in this sentence is misleading, please delete them.
2. While the introduction provides a good overview of biased signaling and the development of biased agonists for opioid receptors, a more detailed discussion of the clinical relevance and potential therapeutic applications of KOR-specific biased agonists would be helpful;
3. Line 113: it should be K2275.40, instead of K2275.40;
4. Line 125: give the RMSDs of Cα to one decimal place;
5. Line 171: the word "slight" in sentence "although there were slight changes in EC₅₀" is not correctly used since some mutants have EC₅₀ changes greater than 1 log.
6. Line 173: the description "K2275.40A and Y3127.34A/F showed significantly reduced arrestin activity for both ligands" is inconsistent with the data. Despite the decrease in E_{max} of arrestin activity, the pEC₅₀ of K2275.40A for nalfurafine increased.
7. Line 248: Please add the reference for "Previous MD simulations...".
8. Line 373: The color for JD_{Tic}-bound KOR should be yellow, instead of magenta.
9. Line 450: It should be "N3227.44 via W2876.48", instead of "T3227.44 via W2876.47".
10. In Supplemental Figure 5, the numbering of K227 is incorrect.

(Remarks to the Author)

NCOMMS-24-16443:

Suno-Ikeda et al. report on the biased signaling properties of two KOR agonists, U50,488H, a balanced KOR agonist and nalfurafine, a Gi-biased KOR agonist. The authors determined crystal structures of KOR with U50,488 and nalfurafine in complex with Gi protein, performed vibrational spectroscopy analysis and carried out cell-based mutational experiments to identify the amino acid residues important for biased signaling of abovementioned ligands. Biased signaling, especially of agonists of opioid ligands, represent an important topic, as they hold promise as next-generation analgesic with improved side effect profile. However, the manuscript at this stage, is not appropriate for publication in Nature Communication. The major points are listed below.

1. The overall interpretation (abstract) that 2 of the 4 identified residues that are pivotal to arrestin recruitment, are also implicated in G protein coupling is not clear: firstly, the authors don't measure G protein coupling, but dissociation of the G protein heterotrimer, and second depending on the ligand all four residues (when mutated) influence potency of G protein dissociation; hence this statement in the abstract needs to be revised.

2. The major concern of this study is the level of novelty. KOR crystal structure with nalfurafine and arrestin-biased agonist WMS-X600, in complex with Nb39 was reported previously by Daibani et al. (<https://www.nature.com/articles/s41467-023-37041-7>). Hence the novelty and information gain of the current study is questionable. In addition, Daibani et al. performed MD simulations of nalfurafine, WMS-X600, and U50,488 bound KOR models, combined with pharmacological assays and provided detailed molecular insights into how KOR agonists induce different conformations of KOR which further results in biased signaling events. Importantly, the present study reports amino acid residues K2275.40, C2866.47, H2916.52, and Y3127.34, as pivotal for KOR biased signaling, whereas two of these (K2275.40; however there is a discrepancy in labelling 5.39 vs. 5.40; Y312,) were reported by Daibani et al. or Che et al. (<https://doi.org/10.1016/j.cell.2017.12.011>); on the other hand, Dalbani et al. and Muratspahic et al. (<https://www.nature.com/articles/s41467-023-43718-w>) report a number of residues that appear to be not relevant in ligand binding and/or ligand bias in the current study (e.g. E209ECL2A, E2976.58A, L3097.32A, see Figure 5, Dalbani et al.); albeit different methods in preparation of the KOR-G protein complex may have been used, these discrepancies need to be carefully studied and explained. More rigorous comparison of the two structures, including control mutational studies (and pharmacological read outs) are required to establish profound reasoning for the reported differences.

3. The study is further limited in that it only considered Gi signaling (and not other G protein isoforms, which is state of the art in high impact manuscripts, e.g. Dalbani et al. and Muratspahic et al.; and as initial read out the current study only uses efficacy measures at one conc., but it would be more relevant to compare potencies.

4. An opioid family wide examination of whether these residues influence ligand bias at MOR/DOR is also missing, which also appears to be state of the art (Dalbani et al.)

5. Pharmacological analyses focus on Gi1 dissociation and arrestin-recruitment assays using different KOR-mutants. However, the authors referred, in several instances, to ligand affinity although they did not report ligand binding data on any of the ligands at the KOR-WT or KOR mutants. Along this line, affinity measurement should be useful to provide further clarification. Some examples are listed below:

- Lines 161-163: "The higher affinity of nalfurafine compared to U-50,488H may be attributed to the greater number of interacting amino acids with nalfurafine." Reference for this statement is missing. Additionally, authors should include ligand binding assays to support this statement and further as comparison between wild-type and mutated constructs to show to which extent mutations actually do affect binding affinity.
- Lines 225-227: "The absence of a hydroxyl group in the Y3127.34 F mutant increases the distance between the phenylalanine side chain and the ligand, weakening the affinity." This statement requires justification since authors did not measure/include data on ligand binding affinity.

6. Ligand bias quantification is absolutely necessary.

7. Some Figure captions are incomplete or lack description of statistical analysis, or use inappropriate statistical tests. Examples are listed below:

- Figure 2: Authors should describe which statistical analysis they used.
- Suppl. Figure 6: Authors should justify using t test, and combining it with one-way ANOVA. Suppl. Fig. 7 should be mentioned before Suppl. Fig. 6, since results shown in Suppl. Fig. 6 are derived from data shown in Suppl. Fig. 7. Same applies to Suppl. Figures 8 and 9.
- Suppl. Figure 10: Q115 residues are mis-labelled.
- Pharmacological data in Figures 3 and 4 are repetitive from data in Suppl. Figures 7 and 9
- Some graphs in Figures 3 and 4 as well as Suppl. Figures 7 and 9 should represent single data points within conc.-response curves.
- In Suppl. Figures 7 and 9 dashed conc.-response curves described in Figure captions do not match the symbol legends presented in figures

8. There is a great deal of discussion included in the results section making it difficult to understand and follow the actual results. The manuscript could be improved if the data were described in more detail in the results section, avoiding too much discussion. For example: lines 207-214, 235-239 and 289-296 should be moved to the discussion section

9. Other comments:

- Lines 61-63: "G protein-coupled receptors (GPCRs) mediate intracellular processes by interacting with signaling transducers such as G proteins and arrestins, a process initiated by binding with extracellular agonists". As much as this is true for the majority of GPCRs, rhodopsin is an exception here by being activated by light and not a diffusible ligand. Please modify accordingly.
- Lines 69-71: "Studies with striatal neurons from arrestin knockout mice have revealed the involvement of the arrestin pathway in mediating adverse effects, including drug aversion and sedation". Authors refer to arrestin knockouts, however reference 6 describes receptor (DOR) knockout animals, and should be replaced accordingly.
- Lines 71-73: "Consequently, there has been a concerted effort to develop G protein-biased agonists with minimal arrestin recruitment activity, aiming to create analgesics devoid of adverse side effects". Definition of side effect per se is an expected and known effect that occurs when the drug is administered. Adverse effect on the other hand is harmful and undesired. Authors should make clear to which one they refer to.
- "G-protein biased" should be consistent with the hyphenation throughout the text (G-protein-biased).
- Line 76 "TRV130, PZM21, and SR-17018" should be specified as MOR agonists.
- Lines 83-87 "Similarly, for KOR, compounds like 6'-GNTI, nalfurafine, novocaine, triazole 1.1 and its derivatives, salvinorin A derivatives, and GR89696 have been developed as G protein biased agonists. While nalfurafine, approved as an antipruritic drug, successfully eliminates the side effect of drug aversion, side effect of sedation persists at doses showing analgesic effects." There are no references included, except reference 12 (for nalfurafine); to be added.
- Lines 96-98: "Comparison with molecular dynamics simulations of KOR in the nalfurafine, U-50,488H and arrestin signalling selective agonist WMS-X600 bound states revealed the presence of three active states." The wording of the sentence should be changed and adjusted for clarity and readability - especially for WMS-X600, which is an arrestin-biased and not an arrestin-selective agonist of KOR.
- Lines 102-104: "However, no comparison between the structures of the G protein-biased and balanced agonist binding states has been conducted, necessitating further studies." Not true: Daibani et al. have identified and compared three active states of KOR bound to nalfurafine, WMS-X600, and a balanced agonist U50,488 via MD simulations. They additionally compared active-state KOR-nalfurafine and KOR-MP1104 structure (MP1104-unselective unbalanced opioid agonist). This should be mentioned accordingly.
- Line 113: it's K2275.40 and not F2275.40
- Lines 170-174: "Overall, neither of the ligands exhibited a significant effect on the Emax in G protein activity, although there were slight changes in EC50 (Fig. 2b, Supplementary Fig. 6, and Supplementary Table 1). However, K227 5.40 A and Y312 7.34 A/F showed significantly reduced arrestin activity for both ligands (Fig. 2c and Supplementary Fig. 8)." This statement does not match the data reported in Figure 2. Firstly, both ligands exhibited significant effects on the Emax for the mutations Y1403.34F and I2946.55A in G-protein dissociation assay. Further, K2275.40A and Y3127.34A/F are not the only mutations showing significant reduction in arrestin recruitment. This part should be described in more detail.
- Lines 215-217: "Next, we investigated the Y3127.34 A and Y3127.34 F mutants, which showed a significant reduction in the Emax of arrestin-recruitment activity (Fig. 3c, Supplementary Figs. 6-9, and Supplementary Table 1)." It is not clear to which ligand are authors referring since in Figure 2c mutation Y3127.34 F did not lead to significant decrease in arrestin-recruitment induced by U50,488H. Authors should additionally refer to Figure 2C.
- Lines 248-250: "Previous MD simulations 249 have indicated that when KOR binds to WMS-X600, an arrestin-biased agonist, the side chain of Q1152.60 orient towards both Y3207.42 and Y661.39." Reference is missing.

Reviewer #4

(Remarks to the Author)

This manuscript by Suno-Ikeda et al. uses cryo-EM and infrared spectroscopy to characterize the binding of two ligands, U-50,488H and nalfurafine, to the kappa opioid receptor at less than 3 angstrom resolution. These two ligands were used because of evidence that U-50,488H is unbiased and nalfurafine is a biased kappa agonist. There was a considerable amount of interesting data demonstrating the interactions with amino acid side chains in the binding pocket of the kappa receptor that was verified and further studied with site directed mutagenesis of important amino acids. This was a very technically challenging and extensive study that provides important information about kappa receptor agonists. This is important because kappa agonists make good analgesics but side effect, particularly dysphoria and sedation, limit their usefulness. If these studies can identify binding interactions that limit -arrestin recruitment without affecting G protein binding, and if biased agonists can reduce kappa receptor mediated side effects, this could be very beneficial for development of novel kappa-directed analgesics. The data in this manuscript was generally well presented with detailed methods and appropriate data analysis and discussion. There were several points that could be clarified and one major issue with respect to the significance of this manuscript. These are described below.

1. The major issue pertains to the use of nalfurafine as the biased agonist. The authors don't provide a reference that nalfurafine is biased, they just say it is. In fact, there are publications that say nalfurafine is G protein biased, unbiased, and -arrestin biased. From all of the data presented in this manuscript, it is unclear that when measuring Gi dissociation or -arrestin recruitment there is a significant difference between the two compounds with respect to the activity in the two assays. Although they identify differences in some mutations, the authors don't point out any differences in activation of the wild type receptor. This is a problem if they are using these two ligands to characterize binding interactions leading to biased signaling. The authors cite a recent publication where a different group used these same ligands to examine the kappa receptor binding pocket using X-ray crystallography and also considered nalfurafine to be biased, but in the current manuscript, for the wild type receptor, biased seems to be about 2 fold. It is not clear that is sufficient to make general statements about biased agonism. That being said, the authors do identify certain residues that seem to be more important

for -arrestin than for G proteins, but there is no clear difference in the way nalfurafine and U-50,488H interact with these residues. In fact, the differences between nalfurafine and U-50,488H are quite minor.

2. In the introduction the authors mention several kappa “biased agonists” but don't give references. Are these more biased than nalfurafine?

3. On page 9 the authors say that nalfurafine binds to K227 but U-50,488H and MP1104 do not and mention Figures 1c and 2a. Neither of these figures show residue K227.

4. On page 14, the authors make the point that mutation of K227 affects arrestin recruitment of nalfurafine but not the G protein response. This is true, but it equally affected U-50,488H binding, so it's not clear how this is relevant to biased signaling. It would have been interesting if this mutation didn't affect the biased agonist nalfurafine and had an effect on U-50,488H, but this was not the case.

5. The mutation of Y312 is also confusing. Y312A leads to decrease in pEC50 for G protein activity and decrease in Emax for arrestin recruitment for both ligands. Y312F affected only the G protein pEC50 of U-50,488H. It's not clear why it then follows that the tyrosine hydroxyl is crucial for nalfurafine-mediated arrestin-recruitment.

6. Page 17 talks about Q115 and mentions Supplementary Figure 10. This figure shows D115.

7. The infrared spectroscopy study seems oddly placed in the middle of a great deal of cryo-EM mutagenesis experiments.

8. The legend to Figure 4f is confusion because there is no orange figure (U-50,488H) in this figure.

9. Table 1 in the Supplementary Material shows all of the binding data. There is no indication of what is significantly different than wild type for any mutation. This is important to know.

Reviewer #5

(Remarks to the Author)

Version 1:

Reviewer comments:

Reviewer #1

(Remarks to the Author)

The author's responses to this reviewer's comments are mostly satisfactory. Still the interpretation of the ATR-FTIR data might be more stringent. The authors tend to interpret the data mostly in qualitative terms. For example, in lines 304-306, a “band-shift” of the Cys S-H signal is mentioned, which is attributed to a change in the H-bonding environment. This could be done in a more quantitative manner. The authors report in the response to the reviewer comments “the infrared absorption band corresponding to the S-H stretching vibration of cysteine undergoes a high-frequency shift. This indicates a weakening of the hydrogen bonding strength of cysteine involved in ligand binding.” However, in the manuscript text they describe the data in more general terms. Why? Furthermore, in lines 307-309 the prominent spectral shift from 1651 cm⁻¹ to 1663-1666 cm⁻¹ is interpreted in terms of alpha helix perturbation. It would be reasonable to provide a more distinctive explanation: What kind of perturbation would be consistent with such spectral shift?

A significant part of the section titled “Quantifying agonist-dependent conformational changes in KOR using infrared spectroscopy” describes the effects of various mutations on KOR structure and function with no relation to FTIR.

Pages 42-43: Please describe the procedure of lipid reconstitution of KOR in more detail, as you did in the response to the reviewer comment. A 1:10 protein-lipid molar ratio is not reasonable. Considering that the lipid makes a bilayer structure, on the membrane surface you have 1 protein molecule per 5 lipid molecules when the cross sectional area of a protein is probably larger than that of 5 lipids.

Reviewer #2

(Remarks to the Author)

The authors have addressed most of my concerns. However, I strongly suggest integrating the data from Supplementary Figure 5 into Figure 1, as it is essential for clarifying the pharmacological profiles of the ligands used in this study. Overall, this work provides valuable insights into the determinants of biased signaling at KOR.

Reviewer #3

(Remarks to the Author)

The manuscript has been greatly improved, but given the modest advance made with respect to previous structural studies (as outlined previously), the abstract (and associated parts of discussion and conclusion) should be carefully rewritten for clarity:

1) "Our findings provide a foundation for targeting specific residues in the KOR ligand-binding pocket to enhance KOR-mediated therapeutic effects while minimizing unwanted side effects.

 previous studies which identified the same residues should be acknowledged

2) "In the G protein dissociation assay, the potency of these mutants was decreased, while their efficacy remained unchanged compared to the wild-type."  The data suggest this is not equally true for all four mutants, and it is also ligand dependent; needs to be more clear

Lastly, the nalfurafine conundrum, i.e. lack of G-protein bias in signaling assays needs to be clarified.

Reviewer #4

(Remarks to the Author)

The authors adequately answered all of my comments.

Reviewer #5

(Remarks to the Author)

Reviewer #6

(Remarks to the Author)

Suno-Ikeda et al. investigate biased signalling within the κ -opioid receptor using a combination of CryoEM spectroscopy, mutational experiments, vibrational spectroscopy analysis and molecular dynamics simulations. The article touches on a relevant subject, and provides a wealth of interesting data. Due to my expertise and the stage of the revision process I will focus my comments on the MD part of the paper. The MD section has been prepared using state-of-the-art protocols, providing sufficient simulation times to rationalize the signalling data. Several issues, however raise concerns:

1. For the simulation was the G-protein maintained or removed?

2. Authors should provide some measurement on the convergence of all simulations (i.e. using RMSD of the receptor backbone or the ligand), furthermore I would recommend that the authors provide some info of the convergence of the conformations of residues which they analyze in Fig 5 (Y7.53 and R3.50)

3. The authors link arrestin recruitment with the formation of specific conformations in the intracellular binding site, however the authors provide no structural rationalization, how the specific mutations alter the conformation of residues studied within Fig 5. An analysis of the impact of the mutations on the behavior of the ligand is lacking. Also, the authors should provide some rationalization of how specific mutations lead to the disappearance of state3.

Lastly, the presentation of the figure is quite confusing, the authors identify 4 specific conformational states, and further subdivide them into 8 states, based on the conformation of R3.50. The authors should depict all states at the panel on the right, clearly stating (according to them) which state is linked to which signalling outcome. If possible I would recommend the authors use a 3D surface to plot their results, as with this they could clearly show the observed 8 conformations.

4. "Compared to the WT, the K2275.40A mutant showed a significant reduction in the alternative conformation" - this cannot be known unless the authors carry out a statistical test

5. Authors should make sure the simulation data is available for revision in a publicly available repository

6. Was the system solvated in any way? I.e. were internal waters modelled? What was the protonation state of the conserved D2.50 residue? In the simulation was the G-protein maintained?

7. It has been often reported that often B-arr recruitment is linked to increased residence of ligands in the binding site (i.e. G-protein biased compounds dissociate faster). It would be very interesting if the researchers could discuss this point using both their current data and conceptual models. Perhaps the highlighted mutations affect mostly B-arr2, as they reduce the affinity of the ligand.

Reviewer #7

(Remarks to the Author)

Version 2:

Reviewer comments:

Reviewer #3

(Remarks to the Author)

all previous comments addressed to satisfaction!

Reviewer #4

(Remarks to the Author)

I have no additional comments. The manuscript is satisfactory to me.

Reviewer #5

(Remarks to the Author)

Reviewer #6

(Remarks to the Author)

The authors have addressed all the comments that I raised and I am happy to recommend the paper for publication

Response to editorial comments:

We would like to sincerely thank the reviewers for their insightful comments and constructive feedback. We have carefully addressed each point raised and provided a detailed, point-by-point response below. The reviewers' comments are shown in black, and our responses are highlighted in blue. We have revised the figure organization based on the reviewers' suggestions for improved clarity. Specifically, the results of the signaling assay have been reorganized into new **Figs. 2 and 3**, along with the Supplementary Figures, combining the previous **Figs. 1 and 2**. The original **Fig. 3** has been divided into the new **Figs. 2 and 3**. Additionally, the previous **Fig. 5** has been moved to the Supplementary Figures, while the new **Fig. 5** now presents results from a newly conducted molecular dynamics simulation. The revisions in the manuscript are highlighted in yellow.

Reviewer #1 (Remarks to the Author):

[Comment 1]

“KOR solubilized in detergent was reconstituted into asolectin liposomes with a 10-fold molar excess.” What detergent was used? Were the liposomes unilamellar made by extrusion or sonication? Was the detergent/lipid molar ratio 10:1?

[Reply]

Thank you for your valuable comments. To a purified KOR solubilized sample at a final concentration of 0.01% DDM, asolectin lipid was added to achieve a molar ratio of 1:10 (KOR:asolectin lipid). The mixture was then rotated at 4°C for 2 hours. Following this, DDM was removed from the KOR, and reconstitution was completed by adding BioBeads and mixing for an additional 2 hours. The asolectin lipid used in the reconstitution had been sonicated to form unilamellar structures before being added to the KOR.

[Comment 2]

Why was 200 mM phosphate buffer used? Together with 140 mM KCl it will produce a very high ionic strength.

[Reply]

Thank you for your valuable comments and suggestions. The buffer-perfused ATR-FTIR measurement used in this study is highly sensitive and capable of detecting even subtle changes in the functional groups of amino acids. However, small variations in ionic strength can lead to swelling or shrinkage of the KOR reconstituted sample on the ATR prism, complicating the detection of structural changes induced by ligand binding. To mitigate this issue, we pre-adjusted the ionic strength to prevent such artifacts from occurring during the measurement.

[Comment 3]

The 0.1 mM concentration of naltrexone is too high, certainly not physiologically relevant. Was the sample flushed with buffer to remove unbound naltrexone before FTIR spectral measurement?

[Reply]

Thank you for your thoughtful question. As you correctly noted, the 0.1 mM concentration is relatively high compared to physiological conditions, and we use this concentration to fully inactivate the KOR under our experimental conditions. To address this, we thoroughly washed-out excess naltrexone with ligand-free buffer immediately before substituting it with various agonists (such as nalfurafine or U-50488H). Following this, we performed the agonist binding experiments, which we believe effectively eliminated any potential artifacts caused by the presence of excess naltrexone.

[Comment 4]

What is “absolute absorbance” and how is it different from just absorbance? What reference (background) was used for the absorbance spectra? Please explain “Corrections were made for protein/lipid contraction.”

[Reply]

Thank you for your thoughtful question. The infrared difference spectra induced by ligand binding, which we use in this study, is technically an infrared absorption difference spectrum. This spectrum is obtained by calculating the difference between the absolute absorption spectra of two different ligands when bound to the KOR. To distinguish between the absolute absorption spectra and the difference absorption spectra, we traditionally refer to the former as the "absolute absorption spectra." We understand how this might be confusing, as you pointed out, so we have revised the terminology as follows.

p.43 Lines 714 – 717:

"The ATR-FTIR difference absorbance spectrum (hereafter referred to as difference spectrum) was obtained by subtracting the absolute absorbance spectrum of the naltrexone-bound form from the absolute absorbance spectrum of the nalfurafine or U-50,488H-bound form."

Additionally, the background spectrum measured when obtaining the absolute absorption spectra of each different agonist-bound form in KOR is measured with no sample on the prism.

Furthermore, the explanation regarding the correction for protein/lipid contraction was insufficient, so we have added detailed corrections as follows.

p.43 Lines 717 – 718:

"Spectral corrections were made for protein swelling/shrinkage, water vapor, CO₂, and contributions from water/buffer components in the spectra."

[Comment 5]

Is there evidence that injection of 0.01 mM Nalfurafine or U-50,488 replaces the receptor-bound naltrexone? If the affinity of naltrexone is higher, in addition to its 10-fold higher concentration, the exchange may be far from complete.

[Reply]

Thank you for your valuable suggestion. We are confident that substitution occurs in both the nalfurafine-bound and U-50488H-bound forms for the following reasons.

First, after replacing naltrexone with nalfurafine in its binding state, we attempted to substitute nalfurafine back with naltrexone. However, no difference spectrum reflecting structural changes was obtained (Fig. 1, left). This result suggests that 0.01 mM of nalfurafine binds strongly to the KOR and cannot be displaced by 0.1 mM of naltrexone. In a similar experiment with U-50488H, we observed a mirror image of the difference spectrum (Fig. 1, right), indicating that 0.01 mM of U-50488H and 0.1 mM of naltrexone can effectively replace each other. Taken together, these results demonstrate that both nalfurafine and U-50488H exhibit a higher affinity for the KOR compared to naltrexone,

and we are confident that both agonists are effectively substituting naltrexone in our experiments.

Based on the results outlined above, it is clear that both nalfurafine and U-50488H have

Figure 1: Ligand binding-dependent conformational changes detected by ATR-FTIR spectroscopy. (a) Red and blue lines correspond to the difference spectra resulting from the replacement of the Naltrexone-bound state with the Nalfurafine-bound state, and from the replacement of the Nalfurafine-bound state back to the Naltrexone-bound state, respectively. (b) Red and blue lines correspond to the difference spectra resulting from the replacement of the Naltrexone-bound state with the U-50488H-bound state, and from the replacement of the U-50488H-bound state back to the Naltrexone-bound state, respectively.

a higher affinity for the KOR compared to naltrexone. We are confident that both agonists effectively substitute for naltrexone in our experiments.

[Comment 6]

Lines 279-280: "...distinct changes in the infrared signals emerged at amide-I of an α -helix (1,700–1,600 cm^{-1})." This reads like the whole 1,700–1,600 cm^{-1} region is the alpha helical region. Consider "...distinct spectral changes in the 1666 cm^{-1} - 1650 cm^{-1} region of the amide I band have been detected and ascribed to conformational changes in the alpha helices of the receptor."

[Reply]

The authors sincerely appreciate the reviewer's suggestion and have made the following revisions to the manuscript.

P.19 Lines 299- 304:

"distinct spectral changes were detected in the 1666-1650 cm^{-1} region of the amide-I band. These changes were attributed to conformational shifts in the receptor's α -helix. In

addition to alterations in the receptor's secondary structure, differences were also observed in the vibrational frequency regions, including the C-N stretching vibration of histidine (1200-1100 cm⁻¹)^(Ref 1) and the S-H stretching vibration of cysteine (2600-2500 cm⁻¹)^(Ref 2-4)."

Ref 1: Noguchi, T., Inoue, Y. & Tang, X-S. Structure of a histidine ligand in the photosynthetic oxygen-evolving complex as studied by light-induced Fourier Transform Infrared difference spectroscopy. *Biochemistry* **38**, 10187-10195 (1999).

Ref 2: Kandori, H., Kinoshita, N., Shichida, Y., Maeda, A., Needleman, R. & Lanyi, J. K. Cysteine S-H as a hydrogen-bonding probe in proteins. *J. Am. Chem. Soc* **120**, 5828-5829 (1998).

[Comment 7]

Lines 270-282: "C-N stretching of histidine imidazole (1,200–1,100 cm⁻¹), and S-H stretching of cysteine side chain (2,600–2,500 cm⁻¹)." Please provide references for these wavenumbers of His and Cys side chains.

[Reply]

Thank you for your helpful comment. We have added the reference as suggested. Thank you again for your input.

P.19 Lines 299- 304:

"distinct spectral changes were detected in the 1666-1650 cm⁻¹ region of the amide-I band. These changes were attributed to conformational shifts in the receptor's α -helix. In addition to alterations in the receptor's secondary structure, differences were also observed in the vibrational frequency regions, including the C-N stretching vibration of histidine (1200-1100 cm⁻¹)^(Ref 1) and the S-H stretching vibration of cysteine (2600-2500 cm⁻¹)^(Ref 2-4)."

Ref 1: Noguchi, T., Inoue, Y. & Tang, X-S. Structure of a histidine ligand in the photosynthetic oxygen-evolving complex as studied by light-induced Fourier Transform Infrared difference spectroscopy. *Biochemistry* **38**, 10187-10195 (1999).

Ref 2: Kandori, H., Kinoshita, N., Shichida, Y., Maeda, A., Needleman, R. & Lanyi, J. K. Cysteine S-H as a hydrogen-bonding probe in proteins. *J. Am. Chem. Soc* **120**, 5828-5829 (1998).

[Comment 8]

Legend to Figure 4: “Signals from nalfurafine and U-50,488H were observed as they interacted with the KOR in the presence of the antagonist naltrexone.” Please explain if both agonists bind to the same site and also if the antagonist binds to same site as the agonists. Do the agonists displace the antagonist? (Is it competitive binding to the same site?)

[Reply]

Thank you for your thoughtful question. As noted in our response to Comment 5, both agonists exhibit a stronger affinity for KOR than naltrexone. Specifically, with nalfurafine, the fact that it cannot be displaced by a tenfold higher concentration of naltrexone suggests that both ligands bind to the same site. In other words, nalfurafine and U-50488H compete with naltrexone for binding at the same site, effectively displacing the antagonist.

[Comment 9]

Lines 283-284: How does a decrease in the absorbance intensity of the Cys side chain at 2550 cm^{-1} indicate a difference in the hydrogen-bonding environment of cysteine?

[Reply]

Thank you for your insightful question. As the binding state transitions from the antagonist to the agonist, the infrared absorption band corresponding to the S-H stretching vibration of cysteine undergoes a high-frequency shift. This indicates a weakening of the hydrogen bonding strength of cysteine involved in ligand binding.

Additionally, as you correctly noted, the apparent decrease in the intensity of the S-H stretching vibration band in the agonist-bound form, on the positive side, can be interpreted as being influenced by the S-H stretching vibration band from the antagonist-bound form on the negative side. This overlap results in an apparent reduction in intensity.

[Comment 10]

Lines 284-287: Can you interpret the possible helix perturbation as judged from the amide I spectral changes, i.e. the shift from 1651 cm^{-1} to $1666\text{-}1663\text{ cm}^{-1}$? Have you considered an alpha-I to alpha-II helix transition?

[Reply]

Thank you for your valuable comment. It is well-established that upon activation by agonist binding, GPCRs undergo perturbations in transmembrane helix 6 (TM6), resulting in outward movement of the cytoplasmic side. In our previous work on the muscarinic receptor, we observed that agonist binding caused a low-frequency shift of the amide I band from 1667 cm^{-1} to 1657 cm^{-1} , which we interpreted as a perturbation of TM6 (Figure 4b, bottom line).

In contrast, with the KOR (kappa opioid receptor), we observed a significant high-frequency shift from 1651 cm^{-1} to 1666 cm^{-1} . This suggests a different type of perturbation in TM6 compared to the muscarinic receptor, or possibly a perturbation in other helices. Additionally, as you noted, the frequency shift may indicate a transition from alpha-I to alpha-II. To clarify which helix is undergoing the change, we plan to use isotopic labeling and other techniques in future studies. Specifically, the three amino acid residues at positions 109 to 111 in TM3 are all threonine, which forms a sequence conducive to adopting a typical alpha-2 helix structure. This region also serves as a ligand-binding site. Therefore, in the case of KOR, agonist binding may induce a perturbation in TM3, and this structural change could be detected in the infrared difference spectrum.

Reviewer #2 (Remarks to the Author):

Major concerns:

[Comment 1]

Calculating the bias factor is a widely used method in pharmacological studies of GPCRs to determine whether a ligand behaves as a biased agonist. Since nalfurafine is not a purely G-protein biased KOR agonist, the authors are encouraged to characterize the bias properties of the compounds used in this study by their established system, using the endogenous ligand dynorphin as the reference.

[Reply]

We thank the reviewer for the helpful suggestion. In response to the comment, we examined the biased activity of nalfurafine and U-50,488H using dynorphin as a reference ligand (Figure R1). Our analysis revealed that the biased signaling activity of both nalfurafine and U-50,488H was marginal (Figure R1C). We have revised the relevant sections of the manuscript to more accurately reflect the biased activity of these agonists (P.8, Lines 127-129).

Although the biased activity of these agonists was weak, we believe this study represents significant progress in understanding KOR signal transduction. In addition to identifying the interaction networks regulating β -arrestin signaling, we have shown that disrupting this network leads to a distinct (alternative) conformational state of KOR, providing a structural framework for biased signaling, as discussed in our response to Reviewer #2-#4.

The key residues involved—K^{5.40}, C^{6.47}, and H^{6.52} (but not Y^{7.34})—are conserved in the δ - and μ -opioid receptors, suggesting that these residues may also play a role in biased signaling in other opioid receptors. As such, we believe this study will remain relevant and appealing to a broad readership interested in biased signaling and opioid receptor research.

Figure. R1 (Supplementary Figure 5 in revised manuscript) Evaluation of biased signaling activity of KOR agonists.

(A–B) G-protein- and β -arrestin-mediated signals were examined with the NanoBiT-G dissociation (A) and NanoBiT- β -arrestin recruitment assay (B), respectively. Dynorphin, an endogenous agonist of KOR, was used as a reference ligand. In the concentration–response curves, symbols and error bars represent the mean and SEM, respectively, of three independent experiments, each performed in duplicate.

(C) Ligand bias calculated from the data shown in (A–B). The $\Delta\Delta\text{Log}(E_{\text{max}}/EC_{50})$ value was used to evaluate ligand bias, as described previously (Kolb et. al., (2024), <https://doi.org/10.1111/bph.15811>). Specifically, the relative activity of nalfurafine or U-50,488H to dynorphin was calculated in each signaling assay, and these relative activities were then compared between assays. Statistical significance was calculated by one sample *t* test (ns, $p > 0.05$; *, $p < 0.05$; **, $p < 0.01$; ***, $p < 0.001$).

This new experiment has been added as Supplementary Figure 5, and the corresponding text has been included as follows.

P.8 Lines 127-132

“In our signaling assay, nalfurafine, previously reported as a G protein-biased ligand, functioned as a balanced agonist, much like U-50,488H. We employed a variety of methods to show that four common amino acids in KOR contribute to signal selectivity in two agonists with different scaffolds. Additionally, MD simulations confirmed that these amino acids are essential for arrestin recruitment activity.”

P.11 Lines 170-171

“Using the NanoBiT assay to assess the bias of the two ligands, we found that both functioned as balanced agonists (Supplementary Fig. 5).”

[Comment 2]

For the nalfurafine-bound KOR-Gi complex, the authors state in line 127 that they obtained four different Gi coupling states of KOR. Do these four coupling models correspond to different KOR conformations when bound to nalfurafine? In a previous study reporting the crystal structure of nalfurafine-bound KOR-Nb39, three active-state KOR conformations were observed during molecular simulations (El Daibani, A., et al. (2023). Nat Commun: 14(1), 1338.). Is there any relationship between the KOR conformations observed in this study and those reported previously?

[Reply]

Thank you for your thoughtful question. There were four states of nalfurafine-bound KORs, but the relative positions of the G protein and the receptor varied. Despite this, the conformations of the receptors were nearly identical, with RMSD values ranging from 0.192 to 0.254Å. Therefore, no significant conformational changes, as previously

reported, were observed, as you correctly noted. In response, the text has been revised from Line 148 (previously Lines 127-130 in the original manuscript) to clarify that the structures of the receptors in each model are almost identical.

The relevant text has been updated as follows.

P.9 Lines 145-149

" When we constructed for all four maps and evaluated the differences, the receptor-only structures showed minimal validation, with RMSD values ranging from 0.19 to 0.25 Å. In contrast, the relative positioning of the G proteins in relation to the receptor varied, suggesting different states of G protein binding to the KOR (Supplementary Fig. 2d)."

[Comment 3]

The legends for the figures presenting pharmacological data, such as Figures 2, 3, Supplemental Figures 7 and 9, are not well-explained. After examining the entire manuscript, it is still unclear what WT (1:1), (1:2), ..., (1:16) mean and why the authors set these references. Furthermore, only the response curves for Gi and arrestin recruitment for some KOR mutants include two WT references, WT (1:1) and (1:16), which is confusing. Please clarify these points.

[Reply]

We thank the reviewer for highlighting this unclear aspect. In mutagenesis studies, it is common for mutations to lead to reduced receptor expression levels. To ensure that any observed decrease in receptor activity is due to the mutation itself rather than lower expression levels, we include data from the wild-type receptor under conditions with similarly reduced expression. As shown in Figure 2b and Supplementary Figure 6, certain cellular responses diminish as receptor expression levels decrease. To clarify this approach, we have revised the figure legend as follows:

(Figures 2, 3, 4, Supplementary Figures 7 and 9)

"Data from the wild type with reduced surface expression serve as a reference point for comparison with mutants exhibiting similarly reduced expression levels. WT (1:2), (1:4), etc., represents plasmid amounts reduced to 1/2, 1/4, and so on, relative to the standard level for transfection."

To address the confusion caused by the dashed lines in each mutant, we have revised the figures to include only conditions with matched expression levels for accurate

comparison. We have also revised the figure captions to provide a clearer description of the displayed conditions. As a result, we have revised Figures 2b, 3a, and 4c.

Figure. R2 Corrections in the figure captions related to Figures 3, 4, Supplementary Figures 7 and 9.

[Comment 4]

In line 202, the authors state that the K227^{5.40}-E297^{6.58} interaction is critical for arrestin activity of KOR. Does the KOR-E297^{6.58}A mutant, which also disrupts this polar contact, exhibit G-protein biased activity when activated by U-50,488H? Additionally, the K227^{5.40}-E297^{6.58} polar interaction is absent in the dynorphin-bound KOR structure (Cell, 186(2), 413-427). Does this mean that the bias determinant K227^{5.40} is ligand-dependent? The authors should discuss this point further.

[Reply]

We thank the reviewer for the valuable comments. Following the reviewer's suggestion, we investigated the KOR agonist-induced responses in the E297^{6.58}A mutant (Fig. R3). Unexpectedly, the substitution of E297^{6.58} with alanine did not result in a significant reduction in either G-protein dissociation or β-arrestin recruitment upon stimulation with nalfurafine and U-50,488H. This suggests that the effect of K227^{5.40} on β-arrestin signaling occurs independently of its interaction with E297^{6.58}.

In the original manuscript, based on the differences in ligand recognition by K227^{5.40} and the K227^{5.40}-E297^{6.58} interaction, we had assumed that nalfurafine and U-50,488H would

regulate β -arrestin activity in a ligand-dependent manner. However, during the revision process, we revised our conclusions and now propose that K227^{5,40} modulates the efficacy of β -arrestin signaling in a ligand-independent manner. These updated interpretations have been reflected in the revised manuscript. (P.14 Lines 212-215).

Figure. R3 (Supplementary Figure 12 in the revised manuscript) Evaluation of the contribution of the E297 residue to signal transduction at KOR.

(A) G-protein- and β -arrestin-mediated responses were examined with the NanoBiT-G-protein dissociation assay and the NanoBiT- β -arrestin recruitment assay, respectively. Concentration–response curves of nalfurafine- and U-50,488H-induced responses were shown. The E_{max} value was obtained by measuring the signal span at the maximum response and normalizing it to the signal span of wild-type KOR (1:1). (B–C) Surface expression of the KOR mutants analyzed by flow cytometry. Data are presented as mean values \pm SEM ($n = 3$; dots). Results for the KOR constructs used for G-protein dissociation (B) β -arrestin recruitment (C) were shown. The amounts of the transfected wild-type KOR plasmid were serially diluted to determine the matched expression condition. The color of the bar graph representing the E297^{6,58}A mutant has been matched to the color of the wild type at equivalent expression levels.

The observation that ionic binding between K227^{5,40} and E297^{6,58} is not crucial for signal selectivity, along with the related discussion, has been incorporated into the text as follows.

P.32 Lines 508-522

" When we investigated the KOR agonist-induced responses in the E297^{6,58}A mutant, we found that substituting E297^{6,58} with alanine did not significantly reduce either G

protein dissociation or β -arrestin recruitment upon stimulation with nalfurafine or U-50,488H (Supplementary Fig. 12). This suggests that the effect of K227^{5.40} on β -arrestin recruitment is independent of its interaction with E297^{6.58}. This finding contradicts the claim by Daibani et al. that the ionic bond between K227^{5.40} and E297^{6.58} plays a critical role in β -arrestin recruitment, and that blocking this bond through K227^{5.40}'s interaction with the ligand is necessary for biased agonism. Additionally, analysis of MD simulations in the WT KOR revealed that the distance between the side chain of K227^{5.40} and nalfurafine predominantly peaked at approximately 4 Å or greater, indicating minimal interaction between them (Supplementary Fig. 20). In other words, we found that both U-50,488H and nalfurafine do not strongly interact with K227^{5.40}, yet K227^{5.40} remains crucial for β -arrestin recruitment. These results suggest that the side chain of K227^{5.40} is essential for β -arrestin recruitment because it occupies the space between the ligand and TM5."

[Comment 5]

Since some KOR mutants influence not only the EC₅₀ but also the E_{max} value of Gi and arrestin signaling, it would be better for the authors to calculate and compare the bias factors of KOR mutants for each ligand. This would demonstrate the mutational effect of each residue on the biased signaling of KOR more directly, especially for those residues emphasized by the authors, such as Q115^{2.60}, K227^{5.40}, Y312^{7.34}, C286^{6.47}, and H291^{6.52}.

[Reply]

Thank you for your valuable comment. Following the reviewer's suggestion, we calculated the relative bias activities of KOR mutants for each ligand (Fig. R4). However, we faced challenges in obtaining reliable results, as the calculated bias activity is significantly affected by variations in receptor expression levels. As shown in Supplementary Figures 8 and 10, the pEC₅₀ values for the G-protein response decrease as wild-type KOR expression levels decrease, whereas the pEC₅₀ values for β -arrestin response remain unchanged. As a result, lower receptor expression tends to lead to β -arrestin bias.

In this context, we observed G protein bias for K227^{5.40}A, C286^{6.47}A, H291^{6.52}A, Y312^{7.34}A, and Y312^{7.34}F upon stimulation with U-50,488H. For nalfurafine stimulation, K227^{5.40}A, H291^{6.52}A, and Y312^{7.34}F also exhibited G protein bias, though the extent of the bias was limited in both conditions. Based on these findings, we concluded that bias activity calculations alone are insufficient to conclude the biased signaling behavior of the mutants. Moreover, we emphasize E_{max} -based assessments for evaluating signal transduction, as the bias factor is more sensitive to changes in EC_{50} than to E_{max} .

Figure. R4 Relative bias activities of the KOR mutants

Relative bias activities were calculated from the pharmacological parameter of the NanoBiT-G-protein dissociation assay and the NanoBiT-β-arrestin recruitment assay. Calculation of the bias activity was performed based on the $\Delta\Delta\text{Log}(E_{max}/EC_{50})$ as described previously (Kolb et. al., (2024), <https://doi.org/10.1111/bph.15811>). Specifically, the relative activity of mutants to wild-type KOR upon nalfurafine or U-50,488H stimulation was calculated in each signaling assay, and these relative activities were then compared between assays. Statistical significance was calculated by one sample *t* test (ns, $p > 0.05$; *, $p < 0.05$; **, $p < 0.01$; ***, $p < 0.001$).

[Comment 6]

Please show the local densities of K227^{5.40} and E297^{6.58} in Figure 2a, Y312^{7.34} and I294^{6.55} in Figure 2d, and C286^{6.47} and T321^{7.43} in Figure 4d.

[Reply]

Thank you for pointing this out. The electron density maps for the side chains of each amino acid are now provided in Figure 2a, Figure 3a, and Supplementary Figure 15.

[Comment 7]

The authors have provided a thorough structural analysis and mutagenesis data. However, it would be valuable to include molecular dynamics simulations or computational modeling to further support the proposed interaction networks and conformational changes associated with biased signaling, such as K227^{5.40}A, Y312^{7.34}A and C286^{6.47}A mutants.

[Reply]

Thank you for your valuable comment. Molecular dynamics simulations were conducted on the wild-type receptor as well as the K227^{5.40}A, Y312^{7.34}A, C286^{6.47}A, and H291^{6.52}A mutants (Fig. 5). The results revealed that each of these mutants is less likely to adopt a conformation that favors β -arrestin activation. These findings are thoroughly discussed in a new section of the manuscript (P. 25, Line 393 – P. 28, Line 440).

Minor:

[Comment 1]

Line 44: the words “responsible for” in this sentence is misleading, please delete them.

[Reply]

Thank you for bringing this to our attention. As you suggested, we have removed it (Line 48).

[Comment 2]

While the introduction provides a good overview of biased signaling and the development of biased agonists for opioid receptors, a more detailed discussion of the clinical relevance and potential therapeutic applications of KOR-specific biased agonists would be helpful.

[Reply]

Thank you for your helpful comment. We have added the reference as follows:
P.6 Lines 89-92

" KOR agonists are crucial for pain relief, especially in the treatment of peripheral neuropathic pain, such as pruritus. However, their use is accompanied by side effects, including drug aversion, depression, sedation, and neuroendocrine disturbances¹³."

[Comment 3]

Line 113: it should be K227^{5.40}, instead of K227^{5.40}.

[Reply]

Thank you for pointing this out. This has been corrected (Line 126).

[Comment 4]

Line 125: give the RMSDs of C α to one decimal place.

[Reply]

Thank you for pointing this out. This has been corrected in the revised manuscript (Line 143).

[Comment 5]

Line 171: the word "slight" in sentence "although there were slight changes in EC₅₀" is not correctly used since some mutants have EC₅₀ changes greater than 1 log.

[Reply]

Thank you for your helpful comment. The sentence has been removed, as the EC₅₀ of Y312^{7.34}A has changed significantly, and similar significant changes were observed in other mutants as well.

[Comment 6]

Line 173: the description "K227^{5.40}A and Y312^{7.34}A/F showed significantly reduced arrestin activity for both ligands" is inconsistent with the data. Despite the decrease in E_{max} of arrestin activity, the pEC₅₀ of K227^{5.40}A for nalfurafine increased.

[Reply]

Thank you for your helpful comment. The description of reduced arrestin activity has been revised to reflect a decrease in the E_{max} of arrestin recruitment activity (P.11 lines 174-178).

" Overall, the G protein activity (E_{max}) of each mutant remained largely unaffected by any ligand (Supplementary Fig. 7, 8, and Supplementary Table 1). However, the K227^{5.40}A and Y312^{7.34}A/F mutants exhibited a significant reduction in E_{max} for β -arrestin-recruitment activity with both ligands (Supplementary Fig. 9, 10, and Supplementary Table 1). "

[Comment 7]

Line 248: Please add the reference for “Previous MD simulations...”.

[Reply]

Thank you for your helpful comment. The reference has been added to the revised manuscript (P.17 Line 262)

[Comment 8]

Line 373: The color for JDTic-bound KOR should be yellow, instead of magenta.

[Reply]

Thank you for your helpful comment. This has been corrected in the revised manuscript (P.25 Line 387)

[Comment 9]

Line 450: It should be “N322^{7.44} via W287^{6.48}”, instead of “T322^{7.44} via W287^{6.47}”.

[Reply]

Thank you for pointing this out. This has been corrected in the revised manuscript (P.35 Line 573).

[Comment 10]

In Supplemental Figure 5, the numbering of K227 is incorrect.

[Reply]

Thank you for pointing this out. This has been corrected in the revised manuscript (Supplementary Fig. 6).

Reviewer #3 (Remarks to the Author):

[Comment 1]

The overall interpretation (abstract) that 2 of the 4 identified residues that are pivotal to arrestin recruitment, are also implicated in G protein coupling is not clear: firstly, the authors don't measure G protein coupling, but dissociation of the G protein heterotrimer, and second depending on the ligand all four residues (when mutated) influence potency of G protein dissociation; hence this statement in the abstract needs to be revised.

[Reply]

Thank you for your valuable feedback. We have revised the abstract as follows:

P.4 lines 58-59

" In the G protein dissociation assay, the potency of these mutants was decreased, while their efficacy remained unchanged compared to the wild-type."

[Comment 2]

The major concern of this study is the level of novelty. KOR crystal structure with nalfurafine and arrestin-biased agonist WMS-X600, in complex with Nb39 was reported previously by Daibani et al. (<https://www.nature.com/articles/s41467-023-37041-7>). Hence the novelty and information gain of the current study is questionable. In addition, Daibani et al. performed MD simulations of nalfurafine, WMS-X600, and U50,488 bound KOR models, combined with pharmacological assays and provided detailed molecular insights into how KOR agonists induce different conformations of KOR which further results in biased signaling events. Importantly, the present study reports amino acid residues K2275.40, C2866.47, H2916.52, and Y3127.34, as pivotal for KOR biased signaling, whereas two of these (K2275.40; however there is a discrepancy in labelling 5.39 vs. 5.40; Y312,) were reported by Daibani et al. or Che et al. (<https://doi.org/10.1016/j.cell.2017.12.011>); on the other hand, Dalbani et al. and

Muratspahic et al. (<https://www.nature.com/articles/s41467-023-43718-w>) report a number of residues that appear to be not relevant in ligand binding and/or ligand bias in the current study (e.g. E209ECL2A, E2976.58A, L3097.32A, see Figure 5, Dalbani et al.); albeit different methods in preparation of the KOR-G protein complex may have been used, these discrepancies need to be carefully studied and explained. More rigorous comparison of the two structures, including control mutational studies (and pharmacological read outs) are required to establish profound reasoning for the reported differences.

[Reply]

Thank you for your valuable comments. Daibani et al. solved the crystal structure of the nalfurafine-bound KOR and performed MD simulations on nalfurafine-, U-50488-, and WMS-X600-bound KOR. In contrast, we determined the structures of the nalfurafine- and U-50488H-bound KOR-G protein complexes using cryo-EM. The novelty of our findings lies in the identification of new amino acids involved in signal selectivity.

As you noted, among the four amino acids we focus on (K227^{5.40}, C286^{6.47}, H291^{6.52}, and Y312^{7.34}), K227^{5.40} and Y312^{7.34} have been discussed previously. We compared the results from these two residues with earlier studies and performed a more detailed analysis.

For K227^{5.40}, Daibani et al. reported that the salt bridge with E297^{6.58} is important for arrestin signaling. Our U-50488-bound KOR structure also confirmed these structural findings. However, our revised experimental results showed that the E297^{6.58}A mutant exhibited the same activity as the wild-type, indicating that the K227^{5.40}-E297^{6.58} salt bridge is not involved in signaling activity, contrary to the findings of Daibani et al.

The following text has been incorporated into the revised manuscript to reflect these findings.

P.33 Lines 508-522

"When we investigated the KOR agonist-induced responses in the E297^{6.58}A mutant, we found that substituting E297^{6.58} with alanine did not significantly reduce either G protein dissociation or β -arrestin recruitment upon stimulation with nalfurafine or U-50,488H (Supplementary Fig. 12). This suggests that the effect of K227^{5.40} on β -arrestin recruitment is independent of its interaction with E297^{6.58}. This finding contradicts the claim by Daibani et al. that the ionic bond between K227^{5.40} and E297^{6.58} plays a

critical role in β -arrestin recruitment, and that blocking this bond through K227^{5.40}'s interaction with the ligand is necessary for biased agonism. Additionally, analysis of MD simulations in the WT KOR revealed that the distance between the side chain of K227^{5.40} and nalfurafine predominantly peaked at approximately 4 Å or greater, indicating minimal interaction between them (Supplementary Fig. 20). In other words, we found that both U-50,488H and nalfurafine do not strongly interact with K227^{5.40}, yet K227^{5.40} remains crucial for β -arrestin recruitment. These results suggest that the side chain of K227^{5.40} is essential for β -arrestin recruitment because it occupies the space between the ligand and TM5."

Muratspahic et al. proposed that Y312^{7.34} might be involved in signal selectivity, but the detailed molecular mechanism remained unclear. In this study, we determined the structural information of two KOR-G_i complexes bound to different agonists. Through a combination of pharmacological analyses, we demonstrated that the distance between the Y312^{7.34} side chain and the ligand is crucial for arrestin recruitment activity. Further experiments revealed that Y312^{7.34} is not essential for ligand binding, providing additional insight into the molecular mechanism underlying this process. This conclusion was supported by structural data from two agonist-bound states and pharmacological analysis of the Y312^{7.34}A/F mutants. Additionally, MD simulations of these four amino acid variants showed that each variant adopted a conformation that was incompatible with arrestin signaling, underscoring the critical role of these residues. The following text has been incorporated into the revised manuscript to reflect this summary.

P.16 Lines 244–260

"The distances from the hydroxyl group on the side chain of Y312^{7.34} to U-50,488H, nalfurafine, and MP1104 were measured at 3.9 Å, 3.3 Å, and 3.6 Å, respectively (Fig. 3b). The Y312^{7.34} mutants exhibited different distances from the ligand, which is thought to weaken the van der Waals interaction. To determine if the changes in signaling activity observed in the Y312^{7.34} mutants were due to differences in ligand binding affinity, we conducted ligand binding experiments with the Y312^{7.34}A/F mutants and the wild-type. Using [³H] U-69,593 as the radioactive ligand, the K_d values for the wild-type, Y312^{7.34}A, and Y312^{7.34}F were calculated as 7.79, 12.1, and 6.29 nM, respectively, indicating that the Y312^{7.34}A mutation slightly reduced affinity (Supplementary Fig. 13a). We then performed competitive binding experiments with nalfurafine and U-50,488H to determine the K_i values. The results consistently showed that nalfurafine

had a higher binding affinity than U-50,488H across all mutants. These binding experiment results suggest that the Y312^{7.34}A/F mutations do not significantly impair binding affinity, indicating that Y312^{7.34} is not essential for ligand binding (Supplementary Fig. 13b). Overall, we concluded that while Y312^{7.34} is not critical for agonist binding, the interaction between the side chain of Y312^{7.34} and the agonist play a key role in β -arrestin recruitment activity."

P. 33 line 525-P.34 line 543

"Structural information and mutational analysis of Y312^{7.34} have provided key insights into the development of biased ligands. Specifically, the Y312^{7.34}F mutant displayed reduced β -arrestin-recruitment activity while preserving strong G protein-coupling activity, highlighting the critical role of the hydroxyl group of tyrosine. The distances from the hydroxyl group on the side chain of Y312^{7.34} to U-50,488H and nalfurafine were measured at 3.9 Å and 3.3 Å, respectively (Fig. 3b). KOR-G_i signaling structures indicate that the interaction between these two agonists and the side chain of Y312^{7.34} involves van der Waals forces, and similar interactions are expected between the Y312^{7.34}F phenylalanine side chain and the agonists.

Compared to nalfurafine, U-50,488H shows a greater distance from the side chain of Y312^{7.34}F, resulting in weaker van der Waals interactions. This is likely why nalfurafine did not impact the G protein signaling activity of the Y312^{7.34}F mutant, while U-50,488H did. In other words, for nalfurafine, the distance from the side chain of Y312^{7.34}F was sufficient to support G protein binding but less favorable for β -arrestin recruitment. In contrast, the Y312^{7.34}A mutant exhibited a significantly increased distance between the side chain and both nalfurafine and U-50,488H, which likely impacted both G protein signaling and β -arrestin-recruitment activities for these ligands. MD simulation results revealed that the Y312^{7.34}A side chain did not interact with nalfurafine, as evidenced by distance histogram peaks of approximately 6 Å and 9 Å (Supplementary Fig. 21). "

We also conducted a pharmacological analysis based on structural information for these residues (I135^{3.29}A, E209^{ECL2}A, E297^{6.58}A, L309^{7.32}A, I316^{7.38}A) that, as noted in previous studies, do not appear to be involved in ligand binding or ligand bias. I135^{3.29}A showed low expression levels and imprecise data, so it was used as a reference. The signaling activity of the E209^{ECL2}A and L309^{7.32}A mutants was comparable to that of the wild type, while the I316^{7.38}A mutant exhibited reduced potency in both G-protein dissociation and β -arrestin recruitment assays.

In the NalA-bound KOR structure, weak interactions were observed between the side chains of L209^{ECL2} and L309^{7.32} with the ligand. Alanine mutations at these residues resulted in increased arrestin signaling, suggesting their involvement in signal selectivity. However, no such interactions were observed in the nalfurafine- and U-50488H-bound structures and the signaling activity of these mutants was equivalent to that of the wild type. This suggests that L209^{ECL2} and L309^{7.32} are not essential for arrestin signaling activity.

Additionally, NalA slightly reduced arrestin pathway activity in the I316^{7.38}A mutant. However, in the structures bound to nalfurafine and U-50488H, the I316^{7.38}A mutant displayed nearly the same efficacy as the wild type, with only a slight decrease in potency. These findings indicate that while I316^{7.38} may influence ligand binding, it does not contribute significantly to the signal selectivity of nalfurafine or U-50488H.

Figure. R5 Evaluation of the contribution of the reported mutants to signal transduction at KOR.

(A–B) Surface expression of the KOR mutants analyzed by flow cytometry. Data are presented as mean values \pm SEM ($n = 3$; dots). Results for the KOR constructs used for G-protein dissociation (A) β -arrestin recruitment (B) were shown. The amounts of the transfected wild-type KOR plasmid were serially diluted to determine the matched expression condition. The color of the bar graph representing the mutants has been matched to the color of the wild-type at equivalent expression levels (C). G-protein- and β -arrestin-mediated responses were examined with the NanoBiT-G-protein dissociation

and the NanoBiT- β -arrestin recruitment assay, respectively. Concentration–response curves of nalfurafine- and U-50,488H-induced responses were shown.

[Comment 3]

The study is further limited in that it only considered G_i signaling (and not other G protein isoforms, which is state of the art in high impact manuscripts, e.g. Dalbani et al. and Muratspahic et al.; and as initial ready out the current study only uses efficacy measures at one conc., but it owul dbe more relevant to compare potencies.

[Reply]

We thank the reviewer for their insightful comments. As highlighted by the reviewer, G-protein subtype selectivity is a critical aspect of signal transduction. In response to the reviewer's suggestion, we investigated the G-protein subtype selectivity of the KOR mutants. Consistent with the G_{i1} dissociation data presented in Supplementary Figure 8, the degree of reduction in the pEC₅₀ value varied depending on the agonist and the specific mutation. However, the overall balance between G-protein subtypes remained largely unchanged across the mutants.

Interestingly, we observed that G_z signaling is regulated in a slightly different manner compared to other G-protein subtypes. A more in-depth exploration of how signaling is regulated among the various G-protein subtypes extends beyond the current scope of our study and will require further investigation. This new experiment has been added as Supplementary Figure 11, and the following text describing this analysis has been included in the revised manuscript.

P.11 Lines 178-183

" In addition to the G_{i1} dissociation reaction shown in Supplementary Fig. 8, we explored differences in signal transduction among the G protein subtypes of the mutants. The decrease in pEC₅₀ values varied depending on the agonist and specific mutation, but the overall balance between the $G_{i/o}$ protein subtypes was consistent across the mutants. Notably, G_z signaling showed variations in E_{max} values for the mutants, especially with U-50,488H (Supplementary Fig. 11). "

Figure. R6 (Supplementary Figure 11 in revised manuscript) G-protein subtype selective effects of agonists and mutations

(A) G-protein- and β -arrestin-mediated signals were examined with the NanoBiT-G-protein dissociation assay and the NanoBiT- β -arrestin recruitment assay, respectively. ($n = 1-2$)

(B) Radar chart showing the influence of each mutation to G-protein subtype selectivity. The pEC_{50} and E_{max} values relative to wild-type KOR were shown for each mutant.

[Comment 4]

An opioid family wide examination of whether these residues influence ligand bias at MOR/DOR is also missing, which also appears to be state of the art (Dalbani et al.)

[Reply]

Thank you for your valuable comments. The following sentence has been added to the revised manuscript.

P.35, Lines 560- 563

" Research on the signal selectivity of MOR and DOR does not address these specific amino acids. The available information suggests that C286^{6.47} is part of the CWxP motif, while H291^{6.52} interacts with the side chain of W287^{6.48} during ligand binding and functions as a component of the toggle switch³⁶."

[Comment 5]

Pharmacological analyses focus on Gi1 dissociation and arrestin-recruitment assays using different KOR-mutants. However, the authors referred, in several instances, to ligand affinity although they did not report ligand binding data on any of the ligands at the KOR-WT or KOR mutants. Along this line, affinity measurement should be useful to provide further clarification. Some examples are listed below:

- Lines 161-163: "The higher affinity of nalfurafine compared to U-50,488H may be attributed to the greater number of interacting amino acids with nalfurafine." Reference for this statement is missing. Additionally, authors should include ligand binding assays to support this statement and further as comparison between wild-type and mutated constructs to show to which extent mutations actually do affect binding affinity.

[Reply]

Thank you for your valuable comments.

Upon incorporating the results from the MD simulations, we observed a discrepancy regarding the interaction involving the side chain of K227^{5.40}. Specifically, while the structural data initially suggested an interaction of K227^{5.40} with nalfurafine, the simulation results indicated that such an interaction occurred minimally.

Consequently, the statement "The higher affinity of nalfurafine compared to U-50,488H may be attributed to the greater number of interacting amino acids with nalfurafine," was inaccurate. We have therefore removed this sentence from the revised manuscript.

We appreciate your understanding and guidance, which has helped us improve the accuracy of our work. We have included results from the MD simulations showing that the side chain of K227^{5.40} interacts minimally with nalfurafine.

P. 32, Lines 515-518

“Additionally, analysis of MD simulations in the WT KOR revealed that the distance between the side chain of K227^{5.40} and nalfurafine predominantly peaked at approximately 4 Å or greater, indicating minimal interaction between them (Supplementary Fig. 20).”

• Lines 225-227: “The absence of a hydroxyl group in the Y3127.34 F mutant increases the distance between the phenylalanine side chain and the ligand, weakening the affinity.” This statement requires justification since authors did not measure/include data on ligand binding affinity.

[Reply]

Thank you for your valuable comments. We conducted ligand binding experiments using the Y3127.34A/F mutants and the wild-type receptor. The results revealed no significant changes in binding affinity for these mutants, suggesting that Y3127.34 is not essential for ligand binding. As a result, we have revised the text in Lines 224–239 to reflect this finding, and we have also updated the related content regarding affinity, previously described in Lines 244–260 and 525–543, to align with this conclusion. *“The distances from the hydroxyl group on the side chain of Y3127.34 to U-50,488H, nalfurafine, and MP1104 were measured at 3.9 Å, 3.3 Å, and 3.6 Å, respectively (Fig. 3b). The Y3127.34 mutants exhibited different distances from the ligand, which is thought to weaken the van der Waals interaction. To determine if the changes in signaling activity observed in the Y3127.34 mutants were due to differences in ligand binding affinity, we conducted ligand binding experiments with the Y3127.34A/F mutants and the wild-type. Using [³H] U-69,593 as the radioactive ligand, the K_d values for the wild-type, Y3127.34A, and Y3127.34F were calculated as 7.79, 12.1, and 6.29 nM, respectively, indicating that the Y3127.34A mutation slightly reduced affinity (Supplementary Fig. 13a). We then performed competitive binding experiments with nalfurafine and U-50,488H to determine the K_i values. The results consistently showed that nalfurafine had a higher binding affinity than U-50,488H across all mutants. These binding experiment results suggest that the Y3127.34A/F mutations do not significantly impair binding affinity, indicating that Y3127.34 is not essential for ligand binding (Supplementary Fig. 13b). Overall, we concluded that while Y3127.34 is not critical for agonist binding, the interaction between the side chain of Y3127.34 and the agonist play a key role in β-arrestin recruitment activity.”*

“Structural information and mutational analysis of Y3127.34 have provided key insights

into the development of biased ligands. Specifically, the Y312^{7.34}F mutant displayed reduced β -arrestin-recruitment activity while preserving strong G protein-coupling activity, highlighting the critical role of the hydroxyl group of tyrosine. The distances from the hydroxyl group on the side chain of Y312^{7.34} to U-50,488H and nalfurafine were measured at 3.9 Å and 3.3 Å, respectively (Fig. 3b). KOR-G_i signaling structures indicate that the interaction between these two agonists and the side chain of Y312^{7.34} involves van der Waals forces, and similar interactions are expected between the Y312^{7.34}F phenylalanine side chain and the agonists.

Compared to nalfurafine, U-50,488H shows a greater distance from the side chain of Y312^{7.34}F, resulting in weaker van der Waals interactions. This is likely why nalfurafine did not impact the G protein signaling activity of the Y312^{7.34}F mutant, while U-50,488H did. In other words, for nalfurafine, the distance from the side chain of Y312^{7.34}F was sufficient to support G protein binding but less favorable for β -arrestin recruitment. In contrast, the Y312^{7.34}A mutant exhibited a significantly increased distance between the side chain and both nalfurafine and U-50,488H, which likely impacted both G protein signaling and β -arrestin-recruitment activities for these ligands. MD simulation results revealed that the Y312^{7.34}A side chain did not interact with nalfurafine, as evidenced by distance histogram peaks of approximately 6 Å and 9 Å (Supplementary Fig. 21). “

[Comment 6]

Ligand bias quantification is absolutely necessary.

[Reply]

We thank the reviewer for the suggestion. We also received similar comments from Reviewers #2 and #4 regarding whether nalfurafine is a G protein-biased agonist. Please refer to our response to **Comment 1 from Reviewer #2** for further details. In summary, our signal assay experiments did not indicate G protein-biased activity for nalfurafine in KOR. Nevertheless, we believe that our identification of the four amino acids critical for KOR arrestin recruitment activity represents a significant advancement in understanding the KOR signaling mechanism.

[Comment 7]

Some Figure captions are incomplete or lack description of statistical analysis or use inappropriate statistical tests. Examples are listed below:

- Figure 2: Authors should describe which statistical analysis they used.

- Suppl. Figure 6: Authors should justify using t test and combining it with one-way ANOVA.

- Suppl. Fig. 7 should be mentioned before Suppl. Fig. 6, since results shown in Suppl. Fig. 6 are derived from data shown in Suppl. Fig. 7. Same applies to Suppl. Figures 8 and 9.

[Reply]

Thank you for your helpful comments. We have incorporated these corrections into the revised manuscript.

- Suppl. Figure 10: Q115 residues are mis-labelled.

[Reply]

Thank you for your helpful comments. We have incorporated these corrections into the revised manuscript.

- Pharmacological data in Figures 3 and 4 are repetitive from data in Suppl. Figures 7 and 9

[Reply]

Thank you for bringing this to our attention. This has been corrected in the revised manuscript.

- Some graphs in Figures 3 and 4 as well as Suppl. Figures 7 and 9 should represent single data points within conc.-response curves.

[Reply]

Thank you for your suggestion. In response to your suggestion, we have adjusted the layout of several figures to display the E_{\max} values alongside the concentration-response curves. Specifically, the bar graphs from Figures 1B and 1C have been reorganized and integrated into Figures 2, 3, Supplementary Figures 7, and 9. Please refer to our response to comments #2-3 for further details.

- In Suppl. Figures 7 and 9 dashed conc.-response curves described in Figure captions do not match the symbol legends presented in figures

Thank you for your comment. We have revised the symbol legends in Supplementary Figures 7 and 9 as follows:

[Comment 8]

There is a great deal of discussion included in the results section making it difficult to understand and follow the actual results. The manuscript could be improved if the data were described in more detail in the results section, avoiding too much discussion. For example: lines 207-214, 235-239 and 289-296 should be moved to the discussion section

[Reply]

Thank you for your valuable comments and suggestions. The sections you highlighted have been removed because we needed to revise the discussion based on new experimental findings.

The content in Lines 207–214 has been omitted because the results from the signal assay of the newly generated E297A mutant were inconsistent with the previous discussion. The content in Lines 235–239, which discussed the difference between biased and balanced ligands, has been removed. Since the bias of nalfurafine was not observed in the new experiment, this section no longer aligns with the experimental results. The discussion in lines 289-296, which was not essential to the focus of the paper, has been excluded. Additionally, since infrared spectroscopy has only been applied to a limited number of GPCRs, providing a more detailed discussion of these findings is challenging at this time.

We hope these revisions enhance the manuscript's clarity and alignment with the new data.

[Comment 9]

Other comments:

- Lines 61-63: “G protein-coupled receptors (GPCRs) mediate intracellular processes by interacting with signaling transducers such as G proteins and arrestins, a process initiated by binding with extracellular agonists”. As much as this is true for the majority of GPCRs, rhodopsin is an exception here by being activated by light and not a diffusible ligand. Please modify accordingly.

[Reply]

Thank you for your suggestions. We have made changes to the text in the revised manuscript.

P.5 Lines 65-68

" *G protein-coupled receptors (GPCRs) regulate intracellular processes through interactions with signaling factors like G proteins and β -arrestins. This process is triggered by the binding of extracellular agonists or, in the case of rhodopsin, by the photoisomerization of retinal^l. "*

- Lines 69-71: "Studies with striatal neurons from arrestin knockout mice have revealed the involvement of the arrestin pathway in mediating adverse effects, including drug aversion and sedation". Authors refer to arrestin knockouts, however reference 6 describes receptor (DOR) knockout animals, and should be replaced accordingly.

[Reply]

Thank you for pointing this out. The citation has been updated in the revised manuscript. (P.5 line 76)

4. Bohn, L. M., Lefkowitz, R. J. & Caron, M. G. Differential mechanisms of morphine antinociceptive tolerance revealed in β arrestin-2 knock-out mice. *Journal of Neuroscience* **22**, 10494–10500 (2002).
5. Ho, J. H. *et al.* G protein signaling-biased agonism at the k-opioid receptor is maintained in striatal neurons. *Sci Signal* **11**, (2018).
6. Pradhan, A. A., Befort, K., Nozaki, C., Gavériaux-Ruff, C. & Kieffer, B. L. The delta opioid receptor: An evolving target for the treatment of brain disorders. *Trends Pharmacol Sci* **32**, 581–590 (2011).

- Lines 71-73: "Consequently, there has been a concerted effort to develop G protein-biased agonists with minimal arrestin recruitment activity, aiming to create analgesics devoid of adverse side effects". Definition of side effect per se is an expected and known effect that occurs when the drug is administered. Adverse effect on the other hand is harmful and undesired. Authors should make clear to which one they refer to.

[Reply]

Thank you for your suggestion. We have revised the text to specifically describe the side effects.

P.5 Line 76-80

" *As a result, extensive research has been dedicated to developing G protein-biased agonists that maintain maximal agonist activity while minimizing β -arrestin binding.*

The goal is to create analgesics that reduce side effects such as respiratory depression, drug dependence, sedation, and convulsions or catalepsy^{7,8}. "

- "G-protein biased" should be consistent with the hyphenation throughout the text (G-protein-biased).

[Reply]

Thank you. This has been corrected in the revised manuscript.

- Line 76 "TRV130, PZM21, and SR-17018" should be specified as MOR agonists.

[Reply]

Thank you for your valuable comments and suggestions. We have revised the sentence to specify that it is MOR-specific.

P.6 Lines 82-83

" MOR-selective G protein-biased agonists, including TRV130, PZM21, and SR-17018, were developed to reduce these side effects⁹. "

- Lines 83-87 "Similarly, for KOR, compounds like 6'-GNTI, nalfurafine, novocaine, triazole 1.1 and its derivatives, salvinorin A derivatives, and GR89696 have been developed as G protein biased agonists. While nalfurafine, approved as an antipruritic drug, successfully eliminates the side effect of drug aversion, side effect of sedation persists at doses showing analgesic effects." There are no references included, except reference 12 (for nalfurafine); to be added.

[Reply]

Thank you for your valuable suggestions. The manuscript has been revised to include the references for each drug. Upon further review, it was determined that Novocaine and GR89696 are not selective ligands, so they have been removed from the revised manuscript.

- Lines 96-98: "Comparison with molecular dynamics simulations of KOR in the nalfurafine, U-50,488H and arrestin signalling selective agonist WMS-X600 bound states revealed the presence of three active states." The wording of the sentence should

be changed and adjusted for clarity and readability - especially for WMS-X600, which is an arrestin-biased and not an arrestin-selective agonist of KOR.

[Reply]

Thank you for your valuable suggestion. 'arrestin signalling selective' has been changed to 'arrestin-biased' in the revised manuscript (Line 106).

- Lines 102-104: "However, no comparison between the structures of the G protein-biased and balanced agonist binding states has been conducted, necessitating further studies." Not true: Daibani et al. have identified and compared three active states of KOR bound to nalfurafine, WMS-X600, and a balanced agonist U50,488 via MD simulations. They additionally compared active-state KOR-nalfurafine and KOR-MP1104 structure (MP1104-unselective unbalanced opioid agonist). This should be mentioned accordingly.

[Reply]

Thank you for your valuable suggestions. Daibani and colleagues compared the structures of KOR in the G protein-biased, balanced, and arrestin-biased ligand binding states using MD simulations. We originally stated that they did not compare the cryo-EM structures of these states; however, the phrasing was unclear. We have revised the sentence to more accurately clarify this distinction.

P.7 Line 115-P.8 Line 118

" Daibani et al. also compared the active structures of nanobody-KOR complexes bound to nalfurafine and MP1104. However, to date, no research has been conducted that compares the active structures of G protein-bound KOR to explore the biased signaling of agonists. "

- Line 113: it's K2275.40 and not F2275.40

[Reply]

Thank you for pointing this out. This has been corrected in the revised manuscript (Line 126).

- Lines 170-174: "Overall, neither of the ligands exhibited a significant effect on the

E_{max} in G protein activity, although there were slight changes in EC_{50} (Fig. 2b, Supplementary Fig. 6, and Supplementary Table 1). However, K227^{5.40}A and Y312^{7.34}A/F showed significantly reduced arrestin activity for both ligands (Fig. 2c and Supplementary Fig. 8).” This statement does not match the data reported in Figure 2. Firstly, both ligands exhibited significant effects on the E_{max} for the mutations Y1403.34F and I2946.55A in G-protein dissociation assay. Further, K227^{5.40}A and Y312^{7.34}A/F are not the only mutations showing significant reduction in arrestin recruitment. This part should be described in more detail.

[Reply]

Thank you for your valuable suggestions. The manuscript has been revised to incorporate the following text.

P.11 Lines 174-178

“ Overall, the G protein activity (E_{max}) of each mutant remained largely unaffected by any ligand (Supplementary Fig. 7, 8, and Supplementary Table 1). However, the K227^{5.40}A and Y312^{7.34}A/F mutants exhibited a significant reduction in E_{max} for β -arrestin-recruitment activity with both ligands (Supplementary Fig. 9, 10, and Supplementary Table 1).”

- Lines 215-217: “Next, we investigated the Y312^{7.34}A and Y312^{7.34}F mutants, which showed a significant reduction in the E_{max} of arrestin-recruitment activity (Fig. 3c, Supplementary Figs. 6-9, and Supplementary Table 1).” It is not clear to which ligand are authors referring since in Figure 2c mutation Y312^{7.34}F did not lead to significant decrease in arrestin-recruitment induced by U50,488H. Authors should additionally refer to Figure 2C.

[Reply]

Thank you for your valuable comments and suggestions. The text has been revised for clarity.

P.15 Lines 235-238

“ Next, we examined the Y312^{7.34}A and Y312^{7.34}F mutants, which demonstrated a significant reduction in the E_{max} of arrestin-recruitment activity when treated with nalfurafine and U-50,488H (Fig. 3a, Supplementary Figs. 7-10, and Supplementary Table 1). ”

• Lines 248-250: “Previous MD simulations 249 have indicated that when KOR binds to WMS-X600, an arrestin-biased agonist, the side chain of Q1152.60 orient towards both Y3207.42 and Y661.39.” Reference is missing.

[Reply]

Thank you for pointing this out. The citation has been added to the revised manuscript (Line 262).

21. El Daibani, A. *et al.* Molecular mechanism of biased signaling at the kappa opioid receptor. *Nat Commun* **14**, 1338 (2023).

Reviewer #4 (Remarks to the Author):

[Comment 1]

The major issue pertains to the use of nalfurafine as the biased agonist. The authors don't provide a reference that nalfurafine is biased, they just say it is. In fact, there are publications that say nalfurafine is G protein biased, unbiased, and b -arrestin biased. From all of the data presented in this manuscript, it is unclear that when measuring Gi dissociation or b-arrestin recruitment there is a significant difference between the two compounds with respect to the activity in the two assays. Although they identify differences in some mutations, the authors don't point out any differences in activation of the wild type receptor. This is a problem if they are using these two ligands to characterize binding interactions leading to biased signaling. The authors cite a recent publication where a different group used these same ligands to examine the kappa receptor binding pocket using X-ray crystallography and also considered nalfurafine to be biased, but in the current manuscript, for the wild type receptor, biased seems to be about 2 fold. It is not clear that is sufficient to make general statements about biased agonism. That being said, the authors do identify certain residues that seem to be more important for b -arrestin than for G proteins, but there is no clear difference in the way nalfurafine and U-50,488H interact with these residues. In fact, the differences between nalfurafine and U-50,488H are quite minor.

[Reply]

We thank the reviewer for the helpful suggestion. We also received similar comments from Reviewers #2 and #3 regarding whether nalfurafine is a G protein-biased agonist. Please refer to our response to **Comment 1 from Reviewer #2** for further details. In summary, our signal assay experiments did not indicate G protein-biased activity for nalfurafine in KOR. Nevertheless, we believe that our identification of the four amino acids critical for KOR arrestin recruitment activity represents a significant advancement in understanding the KOR signaling mechanism.

[Comment 2]

In the introduction the authors mention several kappa “biased agonists” but don’t give references. Are these more biased than nalfurafine?

[Reply]

Thank you for your valuable comments and suggestions. References have been added for each ligand in the revised manuscript. However, since the equilibrium ligands used for comparison vary across different experiments, it was not feasible to determine if they exhibit a higher bias than nalfurafine. Additionally, we identified inaccuracies in the original manuscript regarding some ligands. Novocaine, which is not a KOR agonist, has been removed from the manuscript. Similarly, GR89696 has been excluded due to its minimal bias.

[Comment 3]

On page 9 the authors say that nalfurafine binds to K227 but U-50,488H and MP1104 do not and mention Figures 1c and 2a. Neither of these figures show residue K227.

[Reply]

We appreciate your suggestion. The figures have been updated to include the side chain and the notation of K227.

[Comment 4]

On page 14, the authors make the point that mutation of K227 affects arrestin recruitment of nalfurafine but not the G protein response. This is true, but it equally affected U-50,488H binding, so it’s not clear how this is relevant to biased signaling. It would have been interesting if this mutation didn’t affect the biased agonist nalfurafine and had an effect on U-50,488H, but this was not the case.

We thank the reviewer for their valuable remarks. After careful consideration, we have revised our conclusions regarding the ligand-dependent role of the K227 residue. As the reviewer correctly observed, the K227 mutation impacts β -arrestin recruitment for both agonists, with no ligand-dependent effects. Additionally, our calculation of bias factors revealed that nalfurafine is not a G-protein-biased agonist, as we initially believed. Considering these findings, we have revised the relevant sections of the manuscript to focus more on the signal transduction mechanisms of KOR. These points are further discussed in reviewer responses #2-1 and #2-4.

[Comment 5]

The mutation of Y312 is also confusing. Y312A leads to decrease in pEC50 for G protein activity and decrease in Emax for arrestin recruitment for both ligands. Y312F affected only the G protein pEC50 of U-50,488H. It's not clear why it then follows that the tyrosine hydroxyl is crucial for nalfurafine-mediated arrestin-recruitment.

[Reply]

Thank you for your valuable comments. We propose that the changes observed in the signal assays for the Y312A and Y312F mutants are due to differences in the strength of the interaction between the Y312 side chain and the respective ligands. In these mutants, the distance between Y312 and the ligands is greater than in the wild type, leading to weaker van der Waals interactions. We believe the differences in signal assay results are attributable to Y312A's inability to interact with either ligand, while Y312F maintains a closer proximity to nalfurafine compared to U-50,488H. Specifically, the difference in the hydroxyl group of tyrosine and the phenyl group of phenylalanine influences the distance and, in turn, the signal transduction mechanism. The interaction strength between Y312 and the ligand, along with its influence on the conformation of TM7, was previously suggested to be critical for arrestin recruitment activity. However, ligand-binding experiments conducted in response to the reviewer's comment revealed that the Y312A/F mutations did not affect agonist binding, indicating that Y312 is not essential for ligand binding. The relevant text has been revised as follows:

P.16 Lines 244-260

"The distances from the hydroxyl group on the side chain of Y312^{7,34} to U-50,488H, nalfurafine, and MP1104 were measured at 3.9 Å, 3.3 Å, and 3.6 Å, respectively (Fig. 3b). The Y312^{7,34} mutants exhibited different distances from the ligand, which is thought to weaken the van der Waals interaction. To determine if the changes in signaling

activity observed in the Y312^{7.34} mutants were due to differences in ligand binding affinity, we conducted ligand binding experiments with the Y312^{7.34}A/F mutants and the wild-type. Using [³H] U-69,593 as the radioactive ligand, the K_d values for the wild-type, Y312^{7.34}A, and Y312^{7.34}F were calculated as 7.79, 12.1, and 6.29 nM, respectively, indicating that the Y312^{7.34}A mutation slightly reduced affinity (Supplementary Fig. 13a). We then performed competitive binding experiments with nalfurafine and U-50,488H to determine the K_i values. The results consistently showed that nalfurafine had a higher binding affinity than U-50,488H across all mutants. These binding experiment results suggest that the Y312^{7.34}A/F mutations do not significantly impair binding affinity, indicating that Y312^{7.34} is not essential for ligand binding (Supplementary Fig. 13b). Overall, we concluded that while Y312^{7.34} is not critical for agonist binding, the interaction between the side chain of Y312^{7.34} and the agonist play a key role in β -arrestin recruitment activity."

P.33 Line 525- P.34 Line 543

"Structural information and mutational analysis of Y312^{7.34} have provided key insights into the development of biased ligands. Specifically, the Y312^{7.34}F mutant displayed reduced β -arrestin-recruitment activity while preserving strong G protein-coupling activity, highlighting the critical role of the hydroxyl group of tyrosine. The distances from the hydroxyl group on the side chain of Y312^{7.34} to U-50,488H and nalfurafine were measured at 3.9 Å and 3.3 Å, respectively (Fig. 3b). KOR-G_i signaling structures indicate that the interaction between these two agonists and the side chain of Y312^{7.34} involves van der Waals forces, and similar interactions are expected between the Y312^{7.34}F phenylalanine side chain and the agonists.

Compared to nalfurafine, U-50,488H shows a greater distance from the side chain of Y312^{7.34}F, resulting in weaker van der Waals interactions. This is likely why nalfurafine did not impact the G protein signaling activity of the Y312^{7.34}F mutant, while U-50,488H did. In other words, for nalfurafine, the distance from the side chain of Y312^{7.34}F was sufficient to support G protein binding but less favorable for β -arrestin recruitment. In contrast, the Y312^{7.34}A mutant exhibited a significantly increased distance between the side chain and both nalfurafine and U-50,488H, which likely impacted both G protein signaling and β -arrestin-recruitment activities for these ligands. MD simulation results revealed that the Y312^{7.34}A side chain did not interact with nalfurafine, as evidenced by distance histogram peaks of approximately 6 Å and 9 Å (Supplementary Fig. 21)."

[Comment 6]

Page 17 talks about Q115 and mentions Supplementary Figure 10. This figure shows D115.

[Reply]

Thank you for pointing this out. D115 has been updated to Q115 in the revised manuscript (Supplementary Figure 14).

[Comment 7]

The infrared spectroscopy study seems oddly placed in the middle of a great deal of cryo-EM mutagenesis experiments.

[Reply]

We recognize that readers may find it challenging to understand the connection between the static KOR-G protein structure analysis using cryo-electron microscopy and the dynamic KOR-only analysis using infrared spectroscopy. Therefore, we have added and revised the text as follows.

The authors appreciate the reviewer's comment. To address this, we have revised and expanded the text for clarity, as outlined below.

P.19 Lines 292-297

" The binding of an agonist to KOR may create an environment conducive to the binding of each signaling transduction factor. However, to gain detailed structural insights, it is essential to detect the structural changes that occur prior to G protein and arrestin binding. Therefore, we next investigated the conformational changes that take place upon agonist binding to KORs, before the recruitment of signal transducers, using ATR-FTIR spectroscopy²³⁻²⁵ (Figs. 4a and 4b)."

[Comment 8]

The legend to Figure 4f is confusion because there is no orange figure (U-50,488H) in this figure.

[Reply]

Thank you for your valuable comments and suggestions. To avoid confusion, we have revised the text to clarify the colors of the models in each figure.

[Comment 9]

Table 1 in the Supplementary Material shows all of the binding data. There is no indication of what is significantly different than wild type for any mutation. This is important to know.

[Reply]

We appreciate the reviewer's valuable comment. We have included the statistical analysis information in the table below.

Supplementary Table 1. Pharmacological parameters for the G_i-coupling activity and β-arrestin-2 recruiting activity.

G_i dissociation assay

	Nalfurafine						U-50,488H					
	Emax (%WT)			pEC50			Emax (%WT)			pEC50		
	Mean	SEM		Mean	SEM		Mean	SEM		Mean	SEM	
WT(1:1)	100.0	0.0	-	10.22	0.21	-	100.0	0.0	-	8.69	0.08	-
WT(1:2)	100.0	1.0	-	10.11	0.11	-	100.4	0.7	-	8.63	0.05	-
WT(1:4)	98.3	0.8	-	9.96	0.12	-	97.6	1.6	-	8.42	0.02	-
WT(1:8)	90.5	2.1	-	9.86	0.08	-	93.2	0.8	-	8.14	0.02	-
WT(1:16)	77.8	1.0	-	9.83	0.07	-	82.4	0.1	-	7.92	0.06	-
V108A	102.8	1.1	ns	9.95	0.08	ns	97.6	1.4	ns	8.48	0.09	ns
Y140F	93.1	1.0	*	10.05	0.07	ns	88.6	1.3	**	9.23	0.10	ns
W183A	101.1	2.5	ns	9.62	0.06	ns	97.6	2.0	ns	8.93	0.09	ns
K227A	101.7	0.6	ns	10.08	0.03	ns	101.9	0.4	ns	8.11	0.08	ns
C229A	101.4	0.7	ns	10.07	0.04	ns	101.6	1.5	ns	8.94	0.07	ns
C286A	99.0	1.1	ns	9.74	0.05	ns	96.8	0.4	ns	8.39	0.06	ns
H291A	93.6	0.3	ns	9.33	0.10	**	93.3	1.9	ns	7.36	0.07	***
I294A	92.6	1.5	*	9.32	0.36	**	92.7	1.8	*	7.69	0.28	**
Y312A	95.8	3.8	ns	8.90	0.09	***	100.1	2.4	ns	6.73	0.22	***
Y312F	103.0	1.8	ns	9.56	0.02	*	104.3	1.8	ns	7.46	0.18	***
C315A	103.1	1.1	ns	9.99	0.18	*	102.9	1.3	ns	8.05	0.05	**

β-arrestin 2 recruitment assay

	Nalfurafine						U-50,488H					
	Emax (%WT)			pEC50			Emax (%WT)			pEC50		
	Mean	SEM		Mean	SEM		Mean	SEM		Mean	SEM	
WT(1:1)	100.0	0.0	-	8.73	0.05	-	100.0	0.0	-	7.51	0.12	-
WT(1:2)	101.7	4.2	-	8.69	0.06	-	103.4	4.5	-	7.53	0.14	-
WT(1:4)	104.2	6.8	-	8.73	0.03	-	114.1	4.7	-	7.50	0.09	-
WT(1:8)	100.0	10.2	-	8.80	0.01	-	111.8	13.0	-	7.51	0.13	-
WT(1:16)	105.4	13.4	-	8.86	0.01	-	123.5	16.3	-	7.51	0.13	-
V108A	103.7	4.3	ns	8.60	0.02	ns	124.4	6.7	ns	6.94	0.10	***
Y140F	89.2	3.2	ns	8.63	0.06	ns	95.0	6.0	ns	7.80	0.08	*
W183A	80.9	3.1	*	8.34	0.07	***	107.0	3.2	ns	7.48	0.17	ns
K227A	34.1	3.0	*	9.11	0.01	*	46.0	5.5	*	7.08	0.12	*
C229A	93.2	3.4	ns	8.78	0.04	ns	105.3	4.4	ns	7.57	0.14	ns
C286A	26.9	5.3	**	8.94	0.08	ns	32.4	6.7	*	7.43	0.15	ns
H291A	51.8	4.1	**	7.80	0.08	**	39.2	4.1	**	6.43	0.18	**
I294A	91.5	4.9	ns	7.69	0.01	***	87.3	1.8	ns	6.99	0.06	***
Y312A	28.3	3.2	***	8.33	0.13	ns	41.3	9.8	**	5.79	0.05	***
Y312F	38.7	11.3	**	8.50	0.10	ns	66.1	21.2	ns	5.75	0.10	***
C315A	89.7	5.5	ns	8.64	0.04	ns	104.3	8.0	ns	6.79	0.10	***

Pharmacological parameters for the G_i -coupling activity analyzed by the NanoBiT-G-protein dissociation assay and β -arrestin2-recruiting activity analyzed by the NanoBiT- β -arrestin recruitment assay. Data are presented as mean values \pm SEM ($n = 3-4$; dots). For the individual experiments performed in parallel, data were normalized to the wild-type (WT) KOR (1:1) and presented as E_{max} and ΔpEC_{50} . Statistical analyses were performed using the ordinary one-way ANOVA followed by Dunnett tests with the expression-matched (colored) WT response. *ns*, $p > 0.05$; * $p < 0.05$; ** $p < 0.01$; *** $p < 0.001$.

Other changes

The electron density map of the ligand in Fig. 1c is also shown in Supplementary Fig. 4 and has therefore been omitted.

REVIEWER COMMENTS

Reviewer #1 (Remarks to the Author):

The author's responses to this reviewer's comments are mostly satisfactory. Still the interpretation of the ATR-FTIR data might be more stringent. The authors tend to interpret the data mostly in qualitative terms. For example, in lines 304-306, a "band-shift" of the Cys S-H signal is mentioned, which is attributed to a change in the H-bonding environment. This could be done in a more quantitative manner. The authors report in the response to the reviewer comments "the infrared absorption band corresponding to the S-H stretching vibration of cysteine undergoes a high-frequency shift. This indicates a weakening of the hydrogen bonding strength of cysteine involved in ligand binding." However, in the manuscript text they describe the data in more general terms. Why? Furthermore, in lines 307-309 the prominent spectral shift from 1651 cm⁻¹ to 1663-1666 cm⁻¹ is interpreted in terms of alpha helix perturbation. It would be reasonable to provide a more distinctive explanation: What kind of perturbation would be consistent with such spectral shift?

[Reply]

We sincerely appreciate the reviewer's insightful and constructive comments. In response to the concerns regarding the interpretation of the ATR-FTIR data, particularly the shifts in the cysteine S-H stretching vibration and the amide-I band, we have revised the manuscript text to provide a more specific and mechanistic explanation, as follows:

P.20 Line 314-P.21 Line 321

"The band around 2,550 cm⁻¹, characteristic of the S-H stretching vibration of cysteine, exhibited a band-shift for both ligand-bound forms, indicating a difference in the hydrogen-bonding environment of cysteine (refer to **Fig. 4a**). In the inactive state bound to the antagonist naltrexone (negative side), a consistent band was observed at 2,550 cm⁻¹. In contrast, in the active states induced by the agonists nalfurafine and U-50,488H (positive side), the band shifted up to 2570 and 2565 cm⁻¹, respectively. These upshifts suggest that agonist binding weakens or disrupts the hydrogen bond involving cysteine."

P.21 Lines 321-327

"A similar trend was observed in the protein backbone. As shown in **Fig. 4b**, ligand exchange from naltrexone to nalfurafine or U-50,488H resulted in an upshift of the amide-

I band from 1651 cm^{-1} to 1666 and 1663 cm^{-1} , respectively. This spectral shift is consistent with a perturbation of α -helical hydrogen bonding networks, particularly involving the backbone C=O groups. Such upshifts typically reflect a weakening of intrahelical hydrogen bonds, possibly due to subtle local perturbation or changes in helical packing induced by agonist binding.”

Conversely, in Fig. 4b,

A significant part of the section titled “Quantifying agonist-dependent conformational changes in KOR using infrared spectroscopy” describes the effects of various mutations on KOR structure and function with no relation to FTIR.

[Reply]

We appreciate the reviewer’s insightful comment. We agree that the connection between the ATR-FTIR-derived structural insights and the results of the functional assays using cysteine and histidine mutants was not clearly articulated in the original manuscript. To address, we have revised the manuscript to explicitly link the FTIR observations with the mutational data, as shown below:

P. 22 Lines 334-336

“To elucidate the role of specific amino acid residues (cysteine and histidine) identified by FTIR spectroscopy in ligand-induced conformational changes and their relevance to KOR activation, we conducted functional assays using a series of point mutants.

P. 23 Lines 361-368

The FTIR ligand-induced difference spectra revealed a weakening of the hydrogen bonding of cysteine upon agonist binding (**Fig. 4a**). While our FTIR measurements captured conformational changes upon ligand exchange from naltrexone, the structural comparison shown in **Fig. 4d** was based on the JDtic-bound inactive conformation. Thus, the structural transitions may not be entirely equivalent. Nevertheless, the combined evidence from structural modeling, spectroscopy, and functional assays strongly implicates C286^{6.47} and C315^{7.37} as key residues modulating biased signaling in KOR.”

P.25 Lines 394-398

Notably, W287^{6.48}, a well-characterized toggle switch involved in GPCR activation, is in close spatial proximity to the newly identified residues C286^{6.47}, C315^{7.37}, and H291^{6.52}.

Ligand-induced conformational changes originating from the toggle switch may cause a localized kink in TM6, which likely corresponds to the amide-I band upshift observed in our FTIR spectra.”

Pages 42-43: Please describe the procedure of lipid reconstitution of KOR in more detail, as you did in the response to the reviewer comment. A 1:10 protein-lipid molar ratio is not reasonable. Considering that the lipid makes a bilayer structure, on the membrane surface you have 1 protein molecule per 5 lipid molecules when the cross sectional area of a protein is probably larger than that of 5 lipids.

[Reply]

Thank you for your valuable comment. We acknowledge that, in general, a protein-to-lipid molar ratio of 1:10 may seem insufficient for proper reconstitution, especially considering the bilayer structure of lipids. However, it is known that many membrane proteins, including KOR, retain endogenous lipids during purification, and these lipids cannot be completely removed. We consider that these co-purified lipids contribute to membrane reconstitution together with the added lipids. Moreover, in ATR-FTIR measurements, excessive lipid content can lead to unwanted swelling or contraction of the membrane, which may compromise spectral quality and reproducibility. Therefore, the lipid content was carefully optimized to minimize such effects.

We have revised the “Preparation of proteoliposome of KOR” section as follows:

P. 47 Lines 761-765

Preparation of proteoliposome of KOR

For ATR-FTIR spectroscopy measurement, asolectin lipid pre-sonicated to form unilamellar structures was added to the purified KOR (final DDM concentration: 0.01%) at a molar ratio of 1:10 (KOR:lipid), followed by rotational mixing at 4 °C for 2 hours. Subsequently, BioBeads were added to remove detergent, and the mixture was incubated for another 2 hours to complete the reconstitution process. After removing the Biobeads, the reconstituted KOR was.....

Reviewer #2 (Remarks to the Author):

The authors have addressed most of my concerns. However, I strongly suggest

integrating the data from Supplementary Figure 5 into Figure 1, as it is essential for clarifying the pharmacological profiles of the ligands used in this study. Overall, this work provides valuable insights into the determinants of biased signaling at KOR.

[Reply]

Thank you for your valuable suggestions. As you suggested, we have incorporated the original Supplementary Figure 5 into Figure 1. The corresponding figure legend has also been relocated to accompany Figure 1.

Reviewer #3 (Remarks to the Author):

The manuscript has been greatly improved, but given the modest advance made with respect to previous structural studies (as outlined previously), the abstract (and associated parts of discussion and conclusion) should be carefully rewritten for clarity:

1) "Our findings provide a foundation for targeting specific residues in the KOR ligand-binding pocket to enhance KOR-mediated therapeutic effects while minimizing unwanted side effects.

 previous studies which identified the same residues should be acknowledged

[Reply]

Thank you for your valuable comments and suggestions. We have added a citation regarding the previously reported residues and revised the abstract as follows.

"Our research findings provide a foundation for enhancing KOR-mediated therapeutic effects while minimizing unwanted side effects by targeting specific residues within the KOR ligand-binding pocket, including K227^{5,40} and Y312^{7,34}, which have previously been implicated in biased signaling."

2) "In the G protein dissociation assay, the potency of these mutants was decreased, while their efficacy remained unchanged compared to the wild-type."

 The data suggest this is not equally true for all four mutants, and it is also ligand dependent; needs to be more clear

Lastly, the nalfurafine conundrum, i.e. lack of G-protein bias in signaling assays

needs to be clarified.

[Reply]

Thank you for your valuable comments and suggestions. As you pointed out, we have revised the text to better reflect the content of the main text and have incorporated it into the abstract.

"In the G protein dissociation assay, no change in potency was observed for the K227^{5.40}A and C286^{6.47}A mutants upon addition of nalfurafine. In contrast, a reduction in potency was observed in the other mutants treated with nalfurafine, as well as in all mutants treated with U-50,488H, although efficacy remained nearly equivalent to that of the wild-type receptor."

The issue concerning nalfurafine—specifically, its lack of G protein bias in signal transduction assays—was considered a key point in this study. Our response to this issue is summarized at the beginning of the “Response to Editorial Comments.”

Reviewer #4 (Remarks to the Author):

The authors adequately answered all of my comments.

Reviewer #5 (Remarks to the Author):

Reviewer #6 (Remarks to the Author):

Suno-Ikeda et al. investigate biased signaling within the κ -opioid receptor using a combination of CryoEM spectroscopy, mutational experiments, vibrational spectroscopy analysis and molecular dynamics simulations. The article touches on a relevant subject, and provides a wealth of interesting data. Due to my expertise and the stage of the revision process I will focus my comments on the

MD part of the paper. The MD section has been prepared using state-of-the-art protocols, providing sufficient simulation times to rationalize the signaling data. Several issues, however, raise concerns:

1. For the simulation, was the G-protein maintained or removed

[Reply]

Thank you for the comment. The G protein was not included in the MD simulations. Our aim was to investigate the intrinsic conformational dynamics of the G protein-binding site in its unbound state, enabling comparison between the wild-type and mutant KORs prior to G protein engagement. To clarify this point, we have revised the manuscript as follows:

•P27 Lines 426-428.

... MD simulations for each of the wild-type (WT) and four mutants (K227^{5.40}A, Y312^{7.34}A, C286^{6.47}A, and H291^{6.52}A), all complexed with nalfurafine in the absence of G protein, totaling 15 microseconds

•P.56 Lines 926-928.

The G protein was excluded from the cryo-EM structure to investigate the intrinsic dynamics of the intracellular region.

2. Authors should provide some measurement on the convergence of all simulations (i.e. using RMSD of the receptor backbone or the ligand), furthermore I would recommend that the authors provide some info of the convergence of the conformations of residues which they analyze in Fig 5 (Y7.53 and R3.50)

[Reply]

Thank you for this valuable suggestion. To address this point, we have added RMSD for all heavy atoms of KOR and Nalfurafine, as well as of R156^{3.50} and Y330^{7.53}, across all MD simulations performed in this study. These data are now provided in Supplementary Figures 16-18. This has been noted on P.27 line 429- P.28 line 430.

3-1. *The authors link arrestin recruitment with the formation of specific conformations in the intracellular binding site, however the authors provide no structural rationalization, how the specific mutations alter the conformation of residues studied within Fig 5. An analysis of the impact of the mutations on the behavior of the ligand is lacking. Also, the authors should provide some rationalization of how specific mutations lead to the disappearance of state 3.*

[Reply]

Thank you for this insightful comment. We agreed with your suggestion and performed additional analyses to clarify the effect of the mutations on both receptor conformation and ligand behavior. To address the impact of the mutations, we performed interaction fingerprints analysis across all MD trajectories and added the paragraph related to these results in P.28 Lines 431- 441.

3-2. *Lastly, the presentation of the figure is quite confusing, the authors identify 4 specific conformational states, and further subdivide them into 8 states, based on the conformation of R3.50. The authors should depict all states at the panel on the right, clearly stating (according to them) which state is linked to which signaling outcome. If possible I would recommend the authors use a 3D surface to plot their results, as with this they could clearly show the observed 8 conformations.*

[Reply]

Thank you for your helpful comment. We have revised both the figure and the section “*Conformational landscape changes in K227^{5.40}A, Y312^{7.34}A, C286^{6.47}A, and H291^{6.52}A mutants*” to improve clarity. We now begin by clearly defining six representative conformational states and specify that the canonical and alternative active conformations correspond to State 1 and State 3-A, respectively. The revised figure panel now depicts all relevant states, including the subdivisions based on the orientation of R156^{3.50}. We also added the table summarizing the population of each state across all systems (Fig. 5f). We have incorporated these points into the main text (P. 29 Line 464-P.30 Line 478).

While the main findings remain unchanged, we rewrote the MD simulation section to present the classification and interpretation in a clearer and more accessible manner.

Although we considered using a 3D surface plot, we found that the revised figure sufficiently conveys the diversity of conformational states without adding unnecessary complexity. We appreciate your suggestion, which helped us improve the presentation of our results.

4. *“Compared to the WT, the K2275.40A mutant showed a significant reduction in the alternative conformation” - this cannot be known unless the authors carry out a statistical test*

[Reply]

We appreciate the reviewer’s comment. Since the alternative conformation has not yet been experimentally characterized, it isn't easy to directly validate its population. In addition, a rigorous statistical comparison of conformational populations would require enhanced sampling techniques, which are beyond the scope of the current study. To avoid overinterpretation, we have revised the manuscript to remove the term “significant” and to present the observation more descriptively.

P.30 Lines 479–481.

"Compared to the WT, the K227^{5.40}A mutant tends to exhibit a reduction in the population of state 3-A, corresponding to the alternative conformation with R156^{3.50} facing intracellularly."

5. *Authors should make sure the simulation data is available for revision in a publicly available repository*

[Reply]

Thank you for the suggestion. As requested by the editor, we have uploaded the initial coordinates, simulation input files, and the final output coordinate files as supplementary materials to ensure public accessibility through Google Drive (<https://drive.google.com/drive/folders/1ToPXnmn-ewpEF9qpkBzHrkP1VGxOkUVk>).

6. *Was the system solvated in any way? I.e. were internal waters modelled? What*

was the protonation state of the conserved D2.50 residue? In the simulation was the G-protein maintained?

[Reply]

Thank you for these questions. The simulation system was solvated using the TIP3P water model. The conserved D^{2.50} residue was protonated (ASH form in the Amber force field), as this state is commonly adopted in active GPCRs and has also been used in previous studies. The G protein was not included in the simulation setup, as we aimed to investigate receptor dynamics in the absence of G protein binding. To clarify this point, we have revised the manuscript as follows:

P27 Lines 426-428.

... MD simulations for each of the wild-type (WT) and four mutants (K227^{5.40}A, Y312^{7.34}A, C286^{6.47}A, and H291^{6.52}A), all complexed with nalfurafine in the absence of G protein, totaling 15 microseconds of simulation time.

P.56 Lines 926-928.

The G protein was excluded from the cryo-EM structure to investigate the intrinsic dynamics of the intracellular region.

P.56 Lines 929-930.

The conserved D105^{2.50} was protonated as this state is commonly adopted in active GPCRs.

7. It has been often reported that often B-arr recruitment is linked to increased residence of ligands in the binding site (i.e. G-protein biased compounds dissociate faster). It would be very interesting if the researchers could discuss this point using both their current data and conceptual models. Perhaps the highlighted mutations affect mostly B-arr2, as they reduce the affinity of the ligand.

[Reply]

Thank you for this helpful and insightful comment. As noted in our response to comment #3, we performed the protein-ligand interaction fingerprint analysis to assess the effect of the mutations. The results showed that the interaction frequency between H291^{6.52} and nalfurafine was higher in the WT compared to all four mutants. In contrast, the interaction between ^{7.43} and nalfurafine was more frequent in the mutants than in the

WT. Notably, Y320^{7.42} is located deeper within the ligand-binding pocket than H291^{6.52}, suggesting that in the mutants, nalfurafine adopts a deeper binding pose. This may be associated with increased residence time of the ligand. This observation is consistent with the idea that enhanced ligand residence correlates with increased β -arrestin recruitment, as you pointed out.

Reviewer #7 (Remarks to the Author):
